# Regret Analysis of Posterior Sampling-Based Expected Improvement for Bayesian Optimization

**Shion Takeno**  *takeno.s.mllab.nit@gmail.com*
*Department of Mechanical Systems Engineering, Nagoya University*
*RIKEN Center for Advanced Intelligence Project*

**Yu Inatsu**  *inatsu.yu@nitech.ac.jp*
*Department of Computer Science, Nagoya Institute of Technology*

**Masayuki Karasuyama**  *karasuyama@nitech.ac.jp*
*Department of Computer Science, Nagoya Institute of Technology*

**Ichiro Takeuchi**  *takeuchi.ichiro.n6@f.mail.nagoya-u.ac.jp*
*Department of Mechanical Systems Engineering, Nagoya University*
*RIKEN Center for Advanced Intelligence Project*

**Reviewed on OpenReview:** *https://openreview.net/forum?id=v0s9knY99c*

## Abstract

Bayesian optimization is a powerful tool for optimizing an expensive-to-evaluate black-box function. In particular, the effectiveness of expected improvement (EI) has been demonstrated in a wide range of applications. However, theoretical analyses of EI are limited compared with other theoretically established algorithms. This paper analyzes a randomized variant of EI, which evaluates the EI from the maximum of the posterior sample path. We show that this posterior sampling-based random EI achieves the sublinear Bayesian cumulative regret bounds under the assumption that the black-box function follows a Gaussian process. Finally, we demonstrate the effectiveness of the proposed method through numerical experiments.

## 1 Introduction

Bayesian optimization (BO) (Shahriari et al., 2016) is a powerful tool for optimizing expensive-to-evaluate black-box functions. BO aims for optimization with fewer observations due to the expensive evaluation cost of the objective function. For this purpose, BO sequentially queries a candidate input determined by maximizing an acquisition function (AF) based on some Bayesian model. That is, the queried candidate at $t$-th iteration $\boldsymbol{x}_t$ is chosen as $\boldsymbol{x}_t = \arg\max_{\boldsymbol{x}} \alpha(\boldsymbol{x})$, where $\alpha(\boldsymbol{x})$ is an AF computed based on the Bayesian model. A Gaussian process (GP) model (Rasmussen & Williams, 2005) is typically employed in BO. BO has been applied to various fields, such as materials informatics (Ueno et al., 2016), robotics (Berkenkamp et al., 2023), and hyperparameter tuning (Snoek et al., 2012).

Several widely used BO algorithms are based on an improvement from a reference value, which is often set as the current best observation. The GP-based probability of improvement (GP-PI) algorithm (Kushner, 1964) evaluates the AF defined as the probability that a new observation exceeds the current best observation. However, it is known that GP-PI is too exploitative depending on the reference value (Jones, 2001; Shahriari et al., 2016). Therefore, GP-based expected improvement (GP-EI) (Mockus et al., 1978; Jones et al., 1998), which evaluates the expectation of improvement from the current best observation, has been frequently used. The effectiveness of GP-EI has been repeatedly demonstrated not only in the usual BO (Snoek et al., 2012; Ament et al., 2023) but also in several extended BO settings, such as multi-fidelity BO (Huang et al., 2006), constrained BO (Gardner et al., 2014; Gelbart et al., 2014), multi-objective BO (Emmerich et al., 2011), and

BO with large noises (Letham et al., 2019). However, since the usual GP-EI still depends on the current best observation, which can include noise, its heuristic hyperparameter tuning approach has been discussed; for example, see Section 2.3.2 in (Brochu et al., 2010) and (Lizotte, 2008). Hence, GP-EI is often regarded as a heuristic method with strong empirical performance.

Other lines of study have tackled obtaining theoretical regret bounds (for example, Srinivas et al., 2010; Russo & Van Roy, 2014). Seminal work (Srinivas et al., 2010) showed the cumulative regret upper bound for the GP upper confidence bound (GP-UCB) algorithm. Russo & Van Roy (2014); Chowdhury & Gopalan (2017) derived the cumulative regret upper bounds for the GP-based Thompson sampling algorithm (GP-TS). Furthermore, Scarlett (2018) showed the algorithm-independent regret lower bound and a tighter regret upper bound under several conditions. Recently, Takeno et al. (2024) showed that the variant of GP-PI called GP-PI from the maximum of sample path (GP-PIMS) achieves a sublinear Bayesian cumulative regret (BCR) upper bound. GP-PIMS avoids the problem of selecting the reference value by replacing it with the maximum of the posterior sample path, inspired by GP-TS. Furthermore, a recent study (Iwazaki, 2025) showed a tighter high-probability regret upper bound for the usual GP-UCB.

Several studies (Bull, 2011; Wang & de Freitas, 2014; Nguyen et al., 2017; Berk et al., 2019) have attempted to obtain regret guarantees of GP-EI-based algorithms, in which the reference value is often changed from the current best observation to, for example, the maximum of the posterior mean (Wang & de Freitas, 2014), and the maximum of the posterior sample path (Berk et al., 2019). However, as reported in existing studies (Tran-The et al., 2022; Bedi et al., 2022; Hu et al., 2025), several analyses appear incorrect (See Section 3 and Appendix A for more details). Other theoretical analyses in (Wang & de Freitas, 2014; Tran-The et al., 2022) require rescaling the posterior variance, which can deteriorate the practical optimization performance. Hence, a theoretical analysis for GP-EI-based algorithms without posterior variance rescaling has not been established except for the noiseless case (Bull, 2011).

This paper analyzes the posterior sampling-based GP-EI algorithm inspired by GP-PIMS (Takeno et al., 2024), which is GP-EI with the reference value set by the maximum of the posterior sample path. We refer to this algorithm as GP-EI from the maximum of the sample path (GP-EIMS). We show that GP-EIMS achieves the sublinear BCR under the Bayesian setting, where the objective function follows GPs. Moreover, GP-EIMS does not require rescaling of the posterior variance. Finally, we demonstrate the effectiveness of GP-EIMS, which stems from avoiding posterior variance rescaling, compared with other theoretically guaranteed EI-based methods through numerical experiments.

## 2 Preliminary

**Bayesian Optimization:** We consider the black-box optimization for $f : \mathcal{X} \to \mathbb{R}$ formulated as

$$\boldsymbol{x}^* = \arg\max_{\boldsymbol{x} \in \mathcal{X}} f(\boldsymbol{x}),$$

where $\mathcal{X} \subset \mathbb{R}^d$ is an input domain and $d$ is a dimension. We assume that only observations $y = f(\boldsymbol{x}) + \epsilon$, where $\epsilon$ is noise, can be obtained, but are costly. Thus, our goal is to optimize $f$ with the fewest observations possible. For this goal, BO sequentially performs queries on an input point $\boldsymbol{x}_t$ and observes $y_t = f(\boldsymbol{x}_t) + \epsilon_t$, where $t$ is an iteration. The queried input $\boldsymbol{x}_t$ is chosen as $\boldsymbol{x}_t = \arg\max_{\boldsymbol{x} \in \mathcal{X}} \alpha(\boldsymbol{x})$, where $\alpha(\boldsymbol{x})$ is an AF computed based on the Bayesian model updated by the collected dataset.

**Gaussian Process Model:** In this paper, we assume that $f$ follows a GP with a predefined kernel $k : \mathcal{X} \times \mathcal{X} \to \mathbb{R}$, denoted as $f \sim \mathcal{GP}(0, k)$. We further assume that the observation is contaminated by Gaussian noise; that is, $y_t = f(\boldsymbol{x}_t) + \epsilon_t$, where $\epsilon_t \sim \mathcal{N}(0, \sigma^2)$ with a positive variance $\sigma^2 > 0$. Let $\mathcal{D}_{t-1} = \{(\boldsymbol{x}_i, y_i)\}_{i=1}^{t-1}$ be the training data obtained until the beginning of $t$-th iteration. Then, the posterior distribution $p(f \mid \mathcal{D}_{t-1})$ is a GP again (Rasmussen & Williams, 2005), whose mean and variance can be obtained as follows:

$$\mu_{t-1}(\boldsymbol{x}) = \boldsymbol{k}_{t-1}(\boldsymbol{x})^\top \left( \boldsymbol{K} + \sigma^2 \boldsymbol{I}_{t-1} \right)^{-1} \boldsymbol{y}_{t-1}, \tag{1}$$

$$\sigma_{t-1}^2(\boldsymbol{x}) = k(\boldsymbol{x}, \boldsymbol{x}) - \boldsymbol{k}_{t-1}(\boldsymbol{x})^\top \left( \boldsymbol{K} + \sigma^2 \boldsymbol{I}_{t-1} \right)^{-1} \boldsymbol{k}_{t-1}(\boldsymbol{x}), \tag{2}$$

where $\boldsymbol{k}_{t-1}(\boldsymbol{x}) \coloneqq \big(k(\boldsymbol{x}, \boldsymbol{x}_1), \ldots, k(\boldsymbol{x}, \boldsymbol{x}_{t-1})\big)^\top \in \mathbb{R}^{t-1}$ is the kernel vector, $\boldsymbol{K} \in \mathbb{R}^{(t-1)\times(t-1)}$ is the kernel matrix whose $(i,j)$-element is $k(\boldsymbol{x}_i, \boldsymbol{x}_j)$, $\boldsymbol{I}_{t-1} \in \mathbb{R}^{(t-1)\times(t-1)}$ is the identity matrix, and $\boldsymbol{y}_{t-1} \coloneqq (y_1, \ldots, y_{t-1})^\top \in \mathbb{R}^{t-1}$. Hereafter, we denote that a probability density function (PDF) $p(\cdot|\mathcal{D}_{t-1}) = p_t(\cdot)$, a probability $\Pr(\cdot|\mathcal{D}_{t-1}) = \Pr_t(\cdot)$, and an expectation $\mathbb{E}[\cdot|\mathcal{D}_{t-1}] = \mathbb{E}_t[\cdot]$ for brevity.

Furthermore, for the case of continuous $\mathcal{X}$, we assume the following smoothness condition for GPs:

**Assumption 2.1.** Let $\mathcal{X} \subset [0, r]^d$ be a compact and convex set, where $r > 0$. Assume that the kernel $k$ satisfies the following condition on the derivatives of a sample path $f$. There exists the constants $a \geq 1$ and $b > 0$ such that,

$$\Pr\left(L_j > c\right) \leq a \exp\left(-\left(\frac{c}{b}\right)^2\right), \text{ for } j \in [d], \tag{3}$$

where $L_j = \sup_{\boldsymbol{x} \in \mathcal{X}} \left|\frac{\partial f}{\partial \boldsymbol{x}_j}\right|$ and $[d] = \{1, \ldots, d\}$.

This assumption is commonly used (Srinivas et al., 2010; Kandasamy et al., 2018; Takeno et al., 2023; 2024) and holds at least for the squared exponential (SE) and Matérn-$\nu$ kernels with $\nu > 2$ (Theorem 5 in Ghosal & Roy, 2006; Srinivas et al., 2010).

**Maximum Information Gain:** For regret analysis, we will use the quantity called maximum information gain (MIG) (Srinivas et al., 2010; Vakili et al., 2021b):

**Definition 2.2** (Maximum information gain)**.** Let $f \sim \mathcal{GP}(0, k)$ over $\mathcal{X} \subset [0, r]^d$. Let $A = \{\boldsymbol{a}_i\}_{i=1}^T$, where $\boldsymbol{a}_i \in \mathcal{X}$ for all $i \in [T]$. Let $\boldsymbol{f}_A = \big(f(\boldsymbol{a}_i)\big)_{i=1}^T$, $\boldsymbol{\epsilon}_A = \big(\epsilon_i\big)_{i=1}^T$, where $\epsilon_i \sim \mathcal{N}(0, \sigma^2)$ for all $i \in [T]$, and $\boldsymbol{y}_A = \boldsymbol{f}_A + \boldsymbol{\epsilon}_A \in \mathbb{R}^T$. Then, MIG $\gamma_T$ is defined as follows:

$$\gamma_T \coloneqq \max_A I(\boldsymbol{y}_A; \boldsymbol{f}_A) \text{ such that } |A| = T \text{ and } \boldsymbol{a}_i \in \mathcal{X} \text{ for all } i \in [T], \tag{4}$$

where $I$ is the Shannon mutual information.

For frequently used kernels, MIG is known to be sublinear (Srinivas et al., 2010; Vakili et al., 2021b), for example, $\gamma_T = O\big((\log T)^{d+1}\big)$ for the SE kernels and $\gamma_T = O\big(T^{\frac{d}{2\nu+d}}(\log T)^{\frac{2\nu}{2\nu+d}}\big)$ for the Matérn-$\nu$ kernels.

**Performance Measure:** We evaluate the theoretical performance of BO methods by the BCR (Russo & Van Roy, 2014; Takeno et al., 2023; 2024) defined as

$$\mathrm{BCR}_T = \sum_{t=1}^T \mathbb{E}\left[f(\boldsymbol{x}^*) - f(\boldsymbol{x}_t)\right], \tag{5}$$

where the expectation is taken with all the randomness, that is, $f, (\epsilon_i)_{i\in[T]}$, and the randomness of the algorithm. For example, in GP-TS, GP-PIMS, and GP-EIMS, the randomness of the algorithm implies the posterior sampling. Our goal is to show the sublinearity of the BCR since it implies that the simple regret $\mathbb{E}\left[f(\boldsymbol{x}^*) - f(\hat{\boldsymbol{x}}_T)\right] \leq \mathrm{BCR}_T/T \to 0$ as $T \to \infty$ (Proposition 8 in Russo & Van Roy, 2018), where $\hat{\boldsymbol{x}}_T \coloneqq \arg\max_{\boldsymbol{x} \in \mathcal{X}} \mu_T(\boldsymbol{x})$ is a recommendation point at $T$-th iteration. The Bayesian regret has been analyzed in many existing studies (Russo & Van Roy, 2014; 2018; Kandasamy et al., 2018; Paria et al., 2020; Takeno et al., 2023; 2024). Although the Bayesian regret quantifies the average performance of the algorithm, the Bayesian regret upper bound immediately implies a high probability upper bound due to Markov's inequality. For more details, refer (Sec. 3.1 in Russo & Van Roy, 2014).

**GP-based EI algorithm:** GP-EI is the widely used AF for BO (Mockus et al., 1978; Jones et al., 1998). As the name suggests, the AF of GP-EI given $\mathcal{D}_{t-1}$ at $t$-th iteration is defined as follows:

$$\mathrm{EI}\big(\mu_{t-1}(\boldsymbol{x}), \sigma_{t-1}(\boldsymbol{x}), y_{t-1}^{\max}\big) = \mathbb{E}_t\big[\max\{f(\boldsymbol{x}) - y_{t-1}^{\max}, 0\}\big] \tag{6}$$

$$= \begin{cases} \sigma_{t-1}(\boldsymbol{x})\tau\left(\frac{\mu_{t-1}(\boldsymbol{x}) - y_{t-1}^{\max}}{\sigma_{t-1}(\boldsymbol{x})}\right) & \text{if } \sigma_{t-1}(\boldsymbol{x}) > 0, \\ \max\{\mu_{t-1}(\boldsymbol{x}) - y_{t-1}^{\max}, 0\} & \text{if } \sigma_{t-1}(\boldsymbol{x}) = 0, \end{cases} \tag{7}$$

where $y_{t-1}^{\max} = \max_{i \in [t-1]} y_i$ is the current best observation, $\tau : \mathbb{R} \to \mathbb{R}^+$ is

$$\tau(c) = c\Phi(c) + \phi(c), \tag{8}$$

and $\Phi$ and $\phi$ are the cumulative distribution function and the PDF of the standard Gaussian distribution. Mainly due to the noise included in $y_{t-1}^{\max}$, a regret analysis for the original GP-EI is difficult.

## 3 Literature Review for Regret Analysis of GP-EI

The two most commonly used assumptions for the regret analysis in the BO literature are the Bayesian setting that $f$ follows GPs (Srinivas et al., 2010; Russo & Van Roy, 2014; Takeno et al., 2023; 2024) and the frequentist setting that $f$ belongs to the known reproducing kernel Hilbert space (RKHS) (Srinivas et al., 2010; Chowdhury & Gopalan, 2017; Janz et al., 2020; Iwazaki & Takeno, 2025). Furthermore, the analyzed regret can be categorized as cumulative regret $\sum_{t=1}^{T} f(\boldsymbol{x}^*) - f(\boldsymbol{x}_t)$ and simple regret $f(\boldsymbol{x}^*) - f(\hat{\boldsymbol{x}}_t)$ with some recommended input at $t$-th iteration, for example, $\hat{\boldsymbol{x}}_t \coloneqq \arg\max_{\boldsymbol{x} \in \mathcal{X}} \mu_t(\boldsymbol{x})$. In addition, as we discussed above, we refer to its expected variants as BCR and Bayesian simple regret in the Bayesian setting. Although our main analysis concentrates on the BCR analysis in the Bayesian setting, we review the analyses of GP-EI for both regret definitions in both settings.

### 3.1 Regret Analysis for Noise-Free Setting

Seminal works for the regret analysis of GP-EI concentrated on a noise-free setting where observation noise does not exist. Therefore, there is no need to care about the noise. Vazquez & Bect (2010) show that GP-EI asymptotically converges on the optimum for both Bayesian and frequentist settings regarding the stationary and *non-degeneracy* kernels, such as the Matérn kernel family. Grünewälder et al. (2010) proves that the computationally intractable $T$-step look-ahead variant of GP-EI is near-optimal in the noise-free Bayesian setting under an assumption that the prior mean function and the kernel function are Hölder-continuous. Here, near-optimal means that a regret upper bound matches a regret lower bound except for logarithmic factors. Bull (2011) further shows simple regret upper bounds of GP-EI and modified GP-EI algorithms for the Matérn kernel family in the frequentist setting. In particular, the modified GP-EI is shown to be near-optimal.

### 3.2 Regret Analysis with Rescaling Posterior Variance

Wang & de Freitas (2014) show the high-probability cumulative regret upper bound in the frequentist setting for a case where additional noise contaminates observations, as with our setting. For this analysis, Wang & de Freitas (2014) propose to use $\mu_{t-1}^{\max} = \max_{\boldsymbol{x} \in \mathcal{X}} \mu_{t-1}(\boldsymbol{x})$ instead of $y_{t-1}^{\max}$ and to scale $\sigma_{t-1}(\boldsymbol{x})$ by $\nu_t^{1/2} > 0$ in the AF of GP-EI as follows:

$$\text{EI}\big(\mu_{t-1}(\boldsymbol{x}), \nu_t^{1/2}\sigma_{t-1}(\boldsymbol{x}), \mu_{t-1}^{\max}\big) = \nu_t^{1/2}\sigma_{t-1}(\boldsymbol{x})\tau\left(\frac{\mu_{t-1}(\boldsymbol{x}) - \mu_{t-1}^{\max}}{\nu_t^{1/2}\sigma_{t-1}(\boldsymbol{x})}\right), \tag{9}$$

where $\nu_t = \Theta(\gamma_t)$. Their analysis shows $O(\gamma_T\sqrt{T})$[1] cumulative regret upper bound, which is sublinear, for example, for the SE kernel. However, since $\nu_t \gg 1$ in most cases, the above rescaling strengthens the exploration behavior, and the practical effectiveness can deteriorate. As a result, in the practical scenarios, we must tune $\nu_t$ as a hyperparameter. Note that although Wang & de Freitas (2014) further consider a case where hyperparameters of GPs are unknown, we focus on a case where the hyperparameters of GPs are specified beforehand.

Tran-The et al. (2022) analyze $\text{EI}\big(\mu_{t-1}(\boldsymbol{x}), \nu_t^{1/2}\sigma_{t-1}(\boldsymbol{x}), \tilde{\mu}_{t-1}^{\max}\big)$, which avoids an optimization over (continuous) $\mathcal{X}$ by replacing $\mu_{t-1}^{\max}$ in (Wang & de Freitas, 2014) as $\tilde{\mu}_{t-1}^{\max} = \max_{i \in [t-1]} \mu_{t-1}(\boldsymbol{x}_i)$, in the frequentist setting. However, Tran-The et al. (2022) only analyze the simple regret $f(\boldsymbol{x}^*) - f(\hat{\boldsymbol{x}}_t) = O(\gamma_T/\sqrt{T})$. Therefore, even in the frequentist setting, the cumulative regret upper bound of the GP-EI algorithm with $\tilde{\mu}_{t-1}^{\max}$

---

[1]Although Theorem 1 in (Wang & de Freitas, 2014) states $O(\gamma_T^{3/2}\sqrt{T})$, we believe that their proof suggests $O(\gamma_T\sqrt{T})$.

has not been shown to our knowledge. Furthermore, their algorithm suffers from the same problem as (Wang & de Freitas, 2014), that is, the rescaling of posterior variance.

Although Wang & de Freitas (2014); Tran-The et al. (2022) discuss only the frequentist setting, their derivation can be extended to the Bayesian setting by carefully applying well-known proof techniques mainly from (Russo & Van Roy, 2014; Kandasamy et al., 2018; Takeno et al., 2023). For completeness, we show high probability cumulative regret upper bounds and BCR upper bounds in the Bayesian setting of $\text{EI}\big(\mu_{t-1}(\boldsymbol{x}), \nu_t^{1/2}\sigma_{t-1}(\boldsymbol{x}), \mu_{t-1}^{\max}\big)$ and $\text{EI}\big(\mu_{t-1}(\boldsymbol{x}), \nu_t^{1/2}\sigma_{t-1}(\boldsymbol{x}), \tilde{\mu}_{t-1}^{\max}\big)$ in Appendices B.2 and B.3, respectively. Since our analysis for finite input domains can apply almost similarly to the frequentist setting, the cumulative regret incurred by $\text{EI}\big(\mu_{t-1}(\boldsymbol{x}), \nu_t^{1/2}\sigma_{t-1}(\boldsymbol{x}), \tilde{\mu}_{t-1}^{\max}\big)$ is $O(\gamma_T\sqrt{T})$ in the frequentist setting (see Appendix B.4 for details).

Hu et al. (2025) recently analyze the GP-EI-based algorithm combining $\text{EI}\big(\mu_{t-1}(\boldsymbol{x}), \nu_t^{1/2}\sigma_{t-1}(\boldsymbol{x}), \tilde{\mu}_{t-1}^{\max}\big)$ and a candidate elimination based on a quantity $\text{EI}\big(\tilde{\mu}_{t-1}^{\max}, \nu_t^{1/2}\sigma_{t-1}(\boldsymbol{x}), \mu_{t-1}(\boldsymbol{x})\big)/(T-t)$ called evaluation cost. However, although Remark 1 in (Hu et al., 2025) claim that their $O(\gamma_T\sqrt{T})$ cumulative regret upper bound is tighter than the existing result, $O(\gamma_T\sqrt{T})$ cumulative regret upper bound has already been shown in (Wang & de Freitas, 2014) with the reference value $\mu_{t-1}^{\max}$, as described above. Furthermore, even in the case that the reference value is $\tilde{\mu}_{t-1}^{\max}$, the cumulative regret upper bound $O(\gamma_T\sqrt{T})$ can be obtained without any modification to the GP-EI algorithm (see Appendix B). Hence, the importance of the candidate elimination based on the evaluation cost is ambiguous in the theorem.

### 3.3 Other Related Works

Several other studies analyze GP-EI without rescaling of the posterior variance based on (Nguyen et al., 2017). Nguyen et al. (2017) claims that the cumulative regret upper bound for the usual GP-EI with $y_t^{\max}$ is sublinear in the frequentist setting. However, as shortly discussed in Section 3 of (Tran-The et al., 2022), Remark 1 of (Hu et al., 2025), and Section 1 of (Wang et al., 2025), Lemma 7 in (Nguyen et al., 2017), which claims $\sum_{t=1}^{T} \sigma_t^2(\boldsymbol{x}) = O(\gamma_T)$ for all $\boldsymbol{x} \in \mathcal{X}$, is incorrect. Note that the usual upper bound by MIG is given as $\sum_{t=1}^{T} \sigma_t^2(\boldsymbol{x}_t) = O(\gamma_T)$ (Srinivas et al., 2010). The incorrect derivation in (Nguyen et al., 2017) is the last equality in the proof of Lemma 7. The counter-examples are shown in Appendix A.

At least to our knowledge, Eq (8) in (Berk et al., 2019), page 37 in (Grosnit et al., 2021), Eq. (9.26) in (Bedi et al., 2022), Lemma B.3 in (Marisu & Pun, 2023), and Lemma 2 in (Zhou et al., 2024) in the regret analyses of GP-EI-based algorithms are also incorrect by using a similar result from Lemma 7 in (Nguyen et al., 2017). In addition, this mistake has been used for regret analyses for other AFs, such as Lemma 6 of (Nguyen et al., 2019), Lemma 12 in (Husain et al., 2023).

Exploration enhanced EI ($\text{E}^3\text{I}$) (Berk et al., 2019) is the following GP-EI variants:

$$\text{E}^3\text{I}(\boldsymbol{x}) = \mathbb{E}_{g_t^* \sim p_t(f(\boldsymbol{x}^*))}\left[\sigma_{t-1}(\boldsymbol{x})\tau\left(\frac{\mu_{t-1}(\boldsymbol{x}) - g_t^*}{\sigma_{t-1}(\boldsymbol{x})}\right)\right], \tag{10}$$

where $g_t \sim p_t(f)$ is the sample path from the posterior and $g_t^* = \max_{\boldsymbol{x}\in\mathcal{X}} g_t(\boldsymbol{x})$. The expectation regarding $g_t^* \sim p_t(f(\boldsymbol{x}^*))$ is approximated by Monte Carlo (MC) estimation. However, as we already discussed, their analysis appears incorrect. We will consider GP-EIMS, a similar posterior sampling-based GP-EI algorithm that uses only one posterior sample. In the BCR analyses of GP-EIMS, the probability matching property between $g_t^*$ and $f(\boldsymbol{x}^*)$, that is, the randomness caused by the algorithm relying on only one sample, plays a key role, as with the analyses of GP-TS (Russo & Van Roy, 2014; Kandasamy et al., 2018; Takeno et al., 2024).

Fu et al. (2024) analyze the GP-EI algorithm without the posterior variance scaling for the bilevel optimization problem. However, the derivation from Eq. (D.8) to Eq. (D.9) in (Fu et al., 2024) is not obvious. In Eq. (D.8) in (Fu et al., 2024), the regret upper bound has the multiplicative term $\frac{1}{\tau(\beta_T) - \beta_T}$ with $\beta_T = \Theta(\log T)$. Then, $\tau(\beta_T) - \beta_T = \beta_T\phi(\beta_T)\left(\frac{1}{\beta_T} - \frac{1-\Phi(\beta_T)}{\phi(\beta_T)}\right) = O(\beta_T\phi(\beta_T))$ since the mills ratio $\frac{1-\Phi(c)}{\phi(c)} \to \frac{1}{c}$ as $c \to \infty$ (for example, see Eq. (7) in Gasull & Utzet, 2014) and $\frac{1}{c} - \frac{1-\Phi(c)}{\phi(c)} > 0$

---

**Algorithm 1** GP-EIMS

---

**Require:** Input space $\mathcal{X}$, GP prior $\mu = 0$ and $k$, and initial dataset $\mathcal{D}_0$

 1: **for** $t = 1, \dots$ **do**
 2:     Fit GP to $\mathcal{D}_{t-1}$
 3:     Generate a sample path $g_t \sim p(f|\mathcal{D}_{t-1})$
 4:     $g_t^* \leftarrow \max_{\boldsymbol{x} \in \mathcal{X}} g_t(\boldsymbol{x})$
 5:     $\boldsymbol{x}_t \leftarrow \arg\max_{\boldsymbol{x} \in \mathcal{X}} \mathrm{EI}(\mu_{t-1}(\boldsymbol{x}), \sigma_{t-1}(\boldsymbol{x}), g_t^*)$
 6:     Observe $y_t = f(\boldsymbol{x}_t) + \epsilon_t$ and $\mathcal{D}_t \leftarrow \mathcal{D}_{t-1} \cup (\boldsymbol{x}_t, y_t)$
 7: **end for**

---

for all $c > 0$. Therefore, we can see $\frac{1}{\tau(\beta_T) - \beta_T} = \Omega(1/(\beta_T \phi(\beta_T)))$. Recalling $\beta_T = \Theta(\log T)$, the term $\frac{1}{\tau(\beta_T) - \beta_T} = \Omega(T^{\log T}/\log T)$, which is not sublinear.

Wang et al. (2025) tackle a theoretical analysis of the usual EI with $y_t^{\max}$ without the posterior variance scaling. However, Wang et al. (2025) analyze a quantity $f(\boldsymbol{x}^*) - y_T^{\max}$, which is not the established regret definition, and can be negative. Paritularly when the noise variance $\sigma^2$ is large compared with $f(\boldsymbol{x}^*)$, then $f(\boldsymbol{x}^*) - y_T^{\max}$ can diverge to $-\infty$ rapidly by noise. Therefore, analyzing $f(\boldsymbol{x}^*) - y_T^{\max}$ is not advisable in general.

## 4 GP-EIMS

This section provides the algorithm and the BCR upper bounds of GP-EIMS.

### 4.1 Algorithm

As with GP-PIMS (Takeno et al., 2024), we use the sampled maximum defined as follows:

$$g_t^* = \max_{\boldsymbol{x} \in \mathcal{X}} g_t(\boldsymbol{x}), \tag{11}$$

where $g_t \sim p(f|\mathcal{D}_{t-1})$ is the sample path from the posterior. Then, the AF of GP-EIMS is

$$\mathrm{EI}(\mu_{t-1}(\boldsymbol{x}), \sigma_{t-1}(\boldsymbol{x}), g_t^*) = \mathbb{E}_{f(\boldsymbol{x}) \sim p_t(f(\boldsymbol{x}))} \left[ \max\{f(\boldsymbol{x}) - g_t^*, 0\}|g_t^* \right] \tag{12}$$

$$= \sigma_{t-1}(\boldsymbol{x}) \tau \left( \frac{\mu_{t-1}(\boldsymbol{x}) - g_t^*}{\sigma_{t-1}(\boldsymbol{x})} \right). \tag{13}$$

Note that the posterior variance $\sigma_{t-1}^2(\boldsymbol{x})$ is not rescaled. Finally, an algorithm of GP-EIMS is shown in Algorithm 1. The difference from the usual AF of GP-EI is only changing the reference value as $g_t^*$ from $y_{t-1}^{\max}$.

### 4.2 Regret Analysis

First, we can obtain a general BCR upper bound for arbitrary BO algorithms, as shown in (Takeno et al., 2024):

**Lemma 4.1.** *Let*

$$\eta_t := \frac{g_t^* - \mu_{t-1}(\boldsymbol{x}_t)}{\sigma_{t-1}(\boldsymbol{x}_t)}. \tag{14}$$

*Then, the BCR can be bounded from above as follows:*

$$\mathrm{BCR}_T \leq \sqrt{\mathbb{E}\left[ \sum_{t=1}^{T} \eta_t^2 \mathbb{1}\{\eta_t \geq 0\} \right]} \sqrt{C_1 \gamma_T}, \tag{15}$$

*where $C_1 := 2/\log(1 + \sigma^{-2})$ and the indicator function $\mathbb{1}\{\eta_t \geq 0\} = 1$ if $\eta_t \geq 0$, and $0$ otherwise.*

For completeness, we show the proof in Appendix C.1. Hence, we can concentrate on deriving $\sum_{t=1}^{T} \mathbb{E}\left[\eta_t^2 \mathbb{1}\{\eta_t \geq 0\}\right] = o(T^2/\gamma_T)$ based on the property of GP-EIMS so that the resulting BCR upper bounds are sublinear.

For this purpose, the following lemma plays a key role in our analysis. For more detailed proof, see Appendix C.2.

**Lemma 4.2.** *Fix $\beta > 0$ and $U \leq \max_{\boldsymbol{x} \in \mathcal{X}}\left\{\mu_{t-1}(\boldsymbol{x}) + \beta^{1/2}\sigma_{t-1}(\boldsymbol{x})\right\}$. Suppose that $k(\boldsymbol{x}, \boldsymbol{x}) \leq 1$ for all $\boldsymbol{x} \in \mathcal{X}$. Then, the following inequality holds:*

$$\frac{U - \mu_{t-1}(\boldsymbol{x}_t)}{\sigma_{t-1}(\boldsymbol{x}_t)} \leq \sqrt{\log\left(\frac{\sigma^2 + t - 1}{\sigma^2}\right) + \beta + \sqrt{2\pi\beta}}, \tag{16}$$

*where $\boldsymbol{x}_t = \arg\max_{\boldsymbol{x} \in \mathcal{X}} \mathrm{EI}(\mu_{t-1}(\boldsymbol{x}), \sigma_{t-1}(\boldsymbol{x}), U)$.*

*Short proof.* As with (Wang & de Freitas, 2014), by using the properties of $\tau(\cdot)$ and the posterior variance and the definition of $\boldsymbol{x}_t$, we can obtain

$$|\mu_{t-1}(\boldsymbol{x}_t) - U| \leq \left(\sqrt{\log\left(\frac{\sigma^2 + t - 1}{\sigma^2}\right) - 2\log\left(\tau(-\beta^{1/2})\right) - \log(2\pi)}\right)\sigma_{t-1}(\boldsymbol{x}_t). \tag{17}$$

However, since the AF is defined by $U$ instead of $\max_{\boldsymbol{x} \in \mathcal{X}} \mu_{t-1}(\boldsymbol{x})$ in (Wang & de Freitas, 2014), we must further obtain an upper bound of $-2\log\left(\tau(-\beta^{1/2})\right)$. To obtain the upper bound, we leverage the following useful result:

**Lemma 4.3** (Lemma 1 in (Jang, 2011)). *Q-function $1 - \Phi(c)$ can be bounded from above as follows:*

$$1 - \Phi(c) \leq \frac{1}{\sqrt{2\pi}c}\left(1 - \exp\left(-\sqrt{\frac{\pi}{2}}c\right)\right)\exp\left(-\frac{c^2}{2}\right) \tag{18}$$

$$= \frac{1}{c}\left(1 - \exp\left(-\sqrt{\frac{\pi}{2}}c\right)\right)\phi(c), \tag{19}$$

*for all $c > 0$.*

By using Lemma 4.3 and properties of $\Phi$ and $\phi$, we can obtain the following inequality:

$$\tau(-\beta^{1/2}) = -\beta^{1/2}\Phi(-\beta^{1/2}) + \phi(-\beta^{1/2}) \tag{20}$$

$$= -\beta^{1/2}\left(1 - \Phi(\beta^{1/2})\right) + \phi(\beta^{1/2}) \tag{21}$$

$$\geq -\left(1 - \exp\left(-\sqrt{\frac{\pi}{2}}\beta^{1/2}\right)\right)\phi(\beta^{1/2}) + \phi(\beta^{1/2}) \tag{22}$$

$$= \exp\left(-\sqrt{\frac{\pi}{2}}\beta^{1/2}\right)\phi(\beta^{1/2}). \tag{23}$$

Hence, we see that $-2\log\left(\tau(-\beta^{1/2})\right) \leq \sqrt{2\pi\beta} - 2\log(\phi(\beta^{1/2})) = \sqrt{2\pi\beta} + \beta + \log(2\pi)$. Consequently, by combining the results, we obtain the desired result. $\square$

By leveraging Lemmas 4.1 and 4.2, we show BCR upper bounds for discrete and continuous input domains.

### 4.2.1 BCR Upper Bound for Finite Input Domain

We combine the following lemma from (Srinivas et al., 2010; Takeno et al., 2023) with Lemma 4.2:

**Lemma 4.4** (Lemma 4.1 in (Takeno et al., 2023)). *Suppose that $f$ is a sample path from a GP with zero mean and a predefined kernel $k$, and $\mathcal{X}$ is finite. Pick $\delta \in (0,1)$ and $t \geq 1$. Then, for any given $\mathcal{D}_{t-1}$,*

$$\Pr_t\left(f(\boldsymbol{x}) \leq \mu_{t-1}(\boldsymbol{x}) + \beta^{1/2}(\delta)\sigma_{t-1}(\boldsymbol{x}), \forall \boldsymbol{x} \in \mathcal{X}\right) \geq 1 - \delta, \tag{24}$$

*where $\beta(\delta) = 2\log(|\mathcal{X}|/(2\delta))$.*

Since $f(\boldsymbol{x}^*) \mid \mathcal{D}_{t-1}$ and $g_t^* \mid \mathcal{D}_{t-1}$ are identically distributed, this lemma implies that

$$\Pr\left(g_t^* \leq \max_{\boldsymbol{x} \in \mathcal{X}}\{\mu_{t-1}(\boldsymbol{x}) + \beta^{1/2}(\delta)\sigma_{t-1}(\boldsymbol{x})\}\right) \geq 1 - \delta. \tag{25}$$

Thus, applying Lemma 4.2 by substituting $g_t^*$ to $U$, we can derive the following result:

**Corollary 4.5.** *Assume the same condition as in Lemma 4.4. Suppose that $k(\boldsymbol{x}, \boldsymbol{x}) \leq 1$ for all $\boldsymbol{x} \in \mathcal{X}$. Then, by running GP-EIMS, the following inequality holds with probability at least $1 - \delta$:*

$$\eta_t \leq \sqrt{\log\left(\frac{\sigma^2 + t - 1}{\sigma^2}\right) + \beta(\delta) + \sqrt{2\pi\beta(\delta)}}, \tag{26}$$

*where $\beta(\delta) = 2\log(|\mathcal{X}|/(2\delta))$. Furthermore, we obtain*

$$\mathbb{E}\left[\eta_t^2 \mathbb{1}\{\eta_t \geq 0\}\right] \leq \log\left(\frac{\sigma^2 + t - 1}{\sigma^2}\right) + C_2 + \sqrt{2\pi C_2}, \tag{27}$$

*where $C_2 = 2 + 2\log(|\mathcal{X}|/2)$.*

See Appendix C.3 for the proof. The first inequality is a direct consequence of Lemmas 4.2 and 4.4. The second inequality can be derived similarly to the proof of Lemma 3.2 in (Takeno et al., 2024).

Consequently, from Lemma 4.1 and Corollary 4.5, we can obtain the following BCR upper bound:

**Theorem 4.6.** *Let $f \sim \mathcal{GP}(0, k)$, where the kernel function $k$ is normalized as $k(\boldsymbol{x}, \boldsymbol{x}) \leq 1$, and $\mathcal{X}$ be finite. Then, by running GP-EIMS, the BCR can be bounded from above as follows:*

$$\text{BCR}_T \leq \sqrt{C_1 B_T T \gamma_T}, \tag{28}$$

*where $B_T = \log\left(\frac{\sigma^2 + T - 1}{\sigma^2}\right) + C_2 + \sqrt{2\pi C_2}$, $C_1 := 2/\log(1 + \sigma^{-2})$, and $C_2 = 2 + 2\log(|\mathcal{X}|/2)$.*

See Appendix C.3 for the proof. The BCR upper bound of GP-EIMS for finite input domains is $O(\sqrt{T\gamma_T \log(T|\mathcal{X}|)})$, which is sublinear at least for the SE, linear, and Matérn kernels.

### 4.2.2 BCR Upper Bound for Continuous Input Domain

For continuous input domains, as with the prior works (for example, Srinivas et al., 2010; Kandasamy et al., 2018; Takeno et al., 2023; 2024), we consider a discretized input set $\mathcal{X}_t \subset \mathcal{X}$, which is a finite set with each dimension equally divided into $m_t \geq 1$. Thus, $|\mathcal{X}_t| = m_t^d$. Furthermore, we define the nearest point of $\boldsymbol{x}$ in $\mathcal{X}_t$ as $[\boldsymbol{x}]_t = \arg\min_{\boldsymbol{x}' \in \mathcal{X}_t} \|\boldsymbol{x} - \boldsymbol{x}'\|_1$.

If we consider the AF, $\text{EI}(\mu_{t-1}(\boldsymbol{x}), \sigma_{t-1}(\boldsymbol{x}), \tilde{g}_t^*)$, where $\tilde{g}_t^* := \max_{\boldsymbol{x} \in \mathcal{X}_t} g_t(\boldsymbol{x})$, a sublinear BCR upper bound can be derived in a similar way to the proof of Theorem 4.6. However, $m_t$ for $\mathcal{X}_t$ must depend on several unknown parameters such as $a$ and $b$ in Assumption 2.1. Thus, $m_t$ must be tuned by users if the computation of $\tilde{g}_t^*$ is required in practice. Hence, using $\tilde{g}_t^*$ means that a new tuning parameter $m_t$ is involved, which is not preferable. Therefore, we consider a BCR upper bound for $\text{EI}(\mu_{t-1}(\boldsymbol{x}), \sigma_{t-1}(\boldsymbol{x}), g_t^*)$, which does not require the discretization in the actual algorithm, as with the analyses of GP-PIMS (Takeno et al., 2024).

From Lemma 4.4 and Assumption 2.1, we can obtain the following lemma by appropriately controlling $m_t$:

**Lemma 4.7.** *Suppose Assumption 2.1 holds. Pick $\delta \in (0, 1)$ and $t \geq 1$. Then, the following holds:*

$$\Pr_t\left(\forall \boldsymbol{x} \in \mathcal{X}, f(\boldsymbol{x}) \leq \mu_{t-1}([\boldsymbol{x}]_t) + 2\beta_t^{1/2}(\delta/2)\sigma_{t-1}([\boldsymbol{x}]_t)\right) \geq 1 - \delta, \tag{29}$$

*where $\beta_t(\delta) = 2d\log(m_t) - 2\log(2\delta)$ and $m_t = \max\{2, \lceil bdr\sqrt{(\sigma^2 + t - 1)\log(2ad)/\sigma^2}\rceil\}$.*

See Appendix C.4 for the proof. As with the case of finite input domains, since $f(\boldsymbol{x}^*) \mid \mathcal{D}_{t-1}$ and $g_t^* \mid \mathcal{D}_{t-1}$ are identically distributed, Lemmas 4.2 and 4.7 derive

$$\mathbb{E}\left[\eta_t^2 \mathbb{1}\{\eta_t \geq 0\}\right] \leq \log\left(\frac{\sigma^2 + t}{\sigma^2}\right) + C_T + \sqrt{2\pi C_T}, \tag{30}$$

where $C_T = 8(d \log(m_T) + 1)$.

Consequently, the BCR upper bound can be derived:

**Theorem 4.8.** *Suppose Assumption 2.1 holds and $f \sim \mathcal{GP}(0, k)$, where the kernel function $k$ is normalized as $k(\boldsymbol{x}, \boldsymbol{x}) \leq 1$. Then, by running GP-EIMS, the BCR can be bounded from above as follows:*

$$\text{BCR}_T \leq \sqrt{C_1 B_T T \gamma_T}, \tag{31}$$

*where $C_1 \coloneqq 2/\log(1 + \sigma^{-2})$, $B_T = \log\left(\frac{\sigma^2 + T - 1}{\sigma^2}\right) + C_T + \sqrt{2\pi C_T}$, $C_T = 8(d \log(m_T) + 1)$, and $m_T = \max\left\{2, \left\lceil bdr\sqrt{(\sigma^2 + T - 1)\log(2ad)/\sigma^2} \right\rceil\right\}$.*

See Appendix C.4 for the proof. As with Theorem 4.6, $\text{BCR}_T = O(\sqrt{T\gamma_T \log T})$, which is sublinear for widely used kernels, such as the SE, linear, and Matérn kernels.

## 5 Experiments

We performed numerical experiments using synthetic functions generated from GPs, which match the assumptions for our analysis. We set $\mathcal{X} = \{0.0, 0.1, \ldots, 0.9\}^d$ and $d = 4$. Therefore, $|\mathcal{X}| = 10^4$. We employed the SE kernel $k_{\text{SE}}(\boldsymbol{x}, \boldsymbol{x}') = \exp\left(-\|\boldsymbol{x} - \boldsymbol{x}'\|_2^2/(2\ell^2)\right)$. Furthermore, we actually added observation noise $\epsilon \sim \mathcal{N}(0, \sigma^2)$. We report results changing $\ell \in \{0.1, 0.2\}$ and $\sigma \in \{0.01, 0.1, 1\}$. We fixed the hyperparameters of the GP model, that is $\ell$ and $\sigma$, to the parameters used to generate the synthetic functions and observation noise. We set an initial dataset to data that is closest to $2^d$ data generated randomly as a Sobol sequence.

As baselines, we employed GP-UCB (Srinivas et al., 2010), improved randomized GP-UCB (IRGP-UCB) (Takeno et al., 2023), which uses the random variable $\zeta_t$ instead of $\beta_t$ in GP-UCB, GP-TS (Russo & Van Roy, 2014), max-value entropy search (MES) (Wang & Jegelka, 2017), joint entropy search (JES) (Hvarfner et al., 2022; Tu et al., 2022), GP-PIMS (Takeno et al., 2024), GP-EI (Mockus et al., 1978), E$^3$I (Berk et al., 2019), and GP-EI using the maximum of the posterior mean (Wang & de Freitas, 2014). We denote the method of (Wang & de Freitas, 2014) as GP-EI-$\mu^{\max}$. We set the hyperparameters of GP-UCB, IRGP-UCB, and GP-EI-$\mu^{\max}$ to the theoretically derived parameters for the BCR analyses. Therefore, we set $\beta_t = 2\log(|\mathcal{X}|t^2/\sqrt{2\pi} + 1)$ for GP-UCB and $\zeta_t = 2\log(|\mathcal{X}|/2) + Z_t$ with $Z_t \sim \text{Exp}(\lambda = 1/2)$ for IRGP-UCB (Takeno et al., 2023). Furthermore, we set $\nu_t = 2\log(|\mathcal{X}|t^2/\sqrt{2\pi} + 1)$ for GP-EI-$\mu^{\max}$ (see Theorem B.4). The number of MC samples in MES, JES, and E$^3$I was set to 10. We employed random Fourier feature approximations (Rahimi & Recht, 2008) for the posterior sampling in GP-EIMS, GP-TS, MES, JES, E$^3$I, and GP-PIMS.

Figures 1 and 2 show the simple regret and cumulative regret for the SE kernels. We measured optimization performance by the simple regret $f(\boldsymbol{x}^*) - f(\hat{\boldsymbol{x}}_t)$, where $\hat{\boldsymbol{x}}_t = \arg\max_{\boldsymbol{x} \in \mathcal{X}} \mu_t(\boldsymbol{x})$, and the cumulative regret $\sum_{t=1}^{T} f(\boldsymbol{x}^*) - f(\boldsymbol{x}_t)$. We report the mean and standard error over 16 random trials for the generation of the initial dataset, the synthetic functions, the observation noise, and the algorithm's randomness.

GP-UCB and GP-EI-$\mu^{\max}$ are consistently inferior to other methods. This is because the theoretical values of $\beta_t$ and $\nu_t$ are too large in practice. Indeed, IRGP-UCB, which uses $\zeta_t = \Theta(\log|\mathcal{X}|)$ instead of $\beta_t = \Theta(\log(|\mathcal{X}|t))$, shows better performance than GP-UCB and GP-EI-$\mu^{\max}$. However, IRGP-UCB slightly deteriorates when $\ell = 0.2$, particularly for the cumulative regret, since $\zeta_t$ is still large for the objective function with $\ell = 0.2$ whose local optima are not so many. Note that if $\ell$ becomes small, then the objective function has large variability and more local optima.

GP-TS and JES show superior performance when $\ell = 0.2$. However, when $\ell = 0.1$, GP-TS and JES often deteriorate for both simple and cumulative regret. This tendency has been reported to be caused by over-exploration (Takeno et al., 2023; 2024; Shahriari et al., 2016).

MES and E$^3$I show superior performance in terms of cumulative regret, although their regret upper bounds have not been shown, as discussed in (Takeno et al., 2022) and Sec. 3. However, their simple regrets stagnate in the case of $\ell = 0.2$ and $\sigma = 0.01, 0.1$. Therefore, we can interpret that MES and E$^3$I are slightly too exploitative in these experiments since showing the relatively low cumulative regret and large simple regret

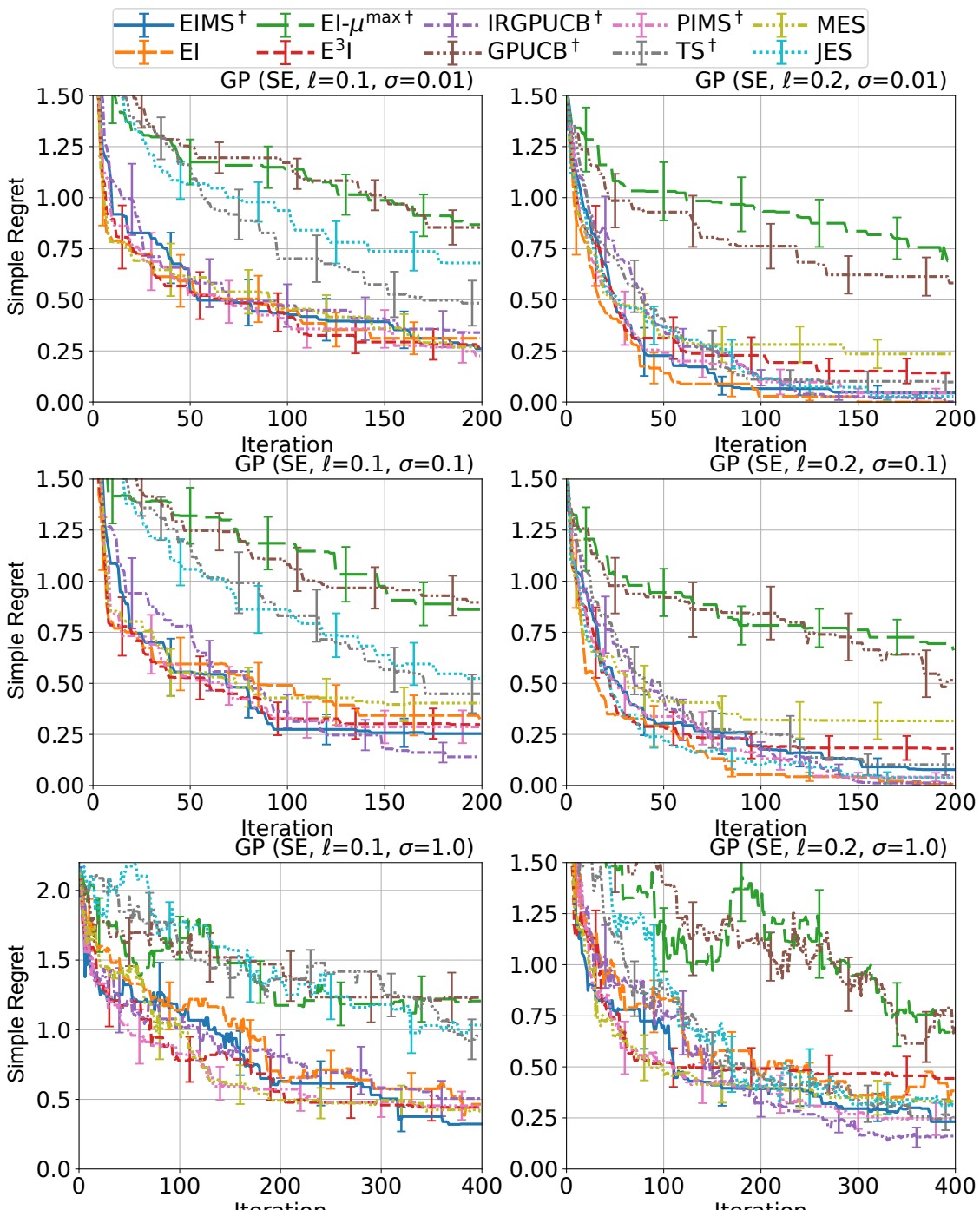

Figure 1: Results of simple regret for synthetic functions generated from a GP defined by the SE kernel. The top, middle, and bottom rows show the result for noise standard deviations $\sigma = 0.01, 0.1$, and 1, respectively. The left and right columns show the result for length scales of the kernel $\ell = 0.1$ and $\ell = 0.2$, respectively. Daggers in the legend indicate that the BO method was performed with theoretical settings, for example, regarding the hyperparameters.

implies that the corresponding BO method repeatedly evaluates non-optimal points whose instantaneous regret is low but not zero. On the other hand, it is known that GP-PIMS can be viewed as MES using one

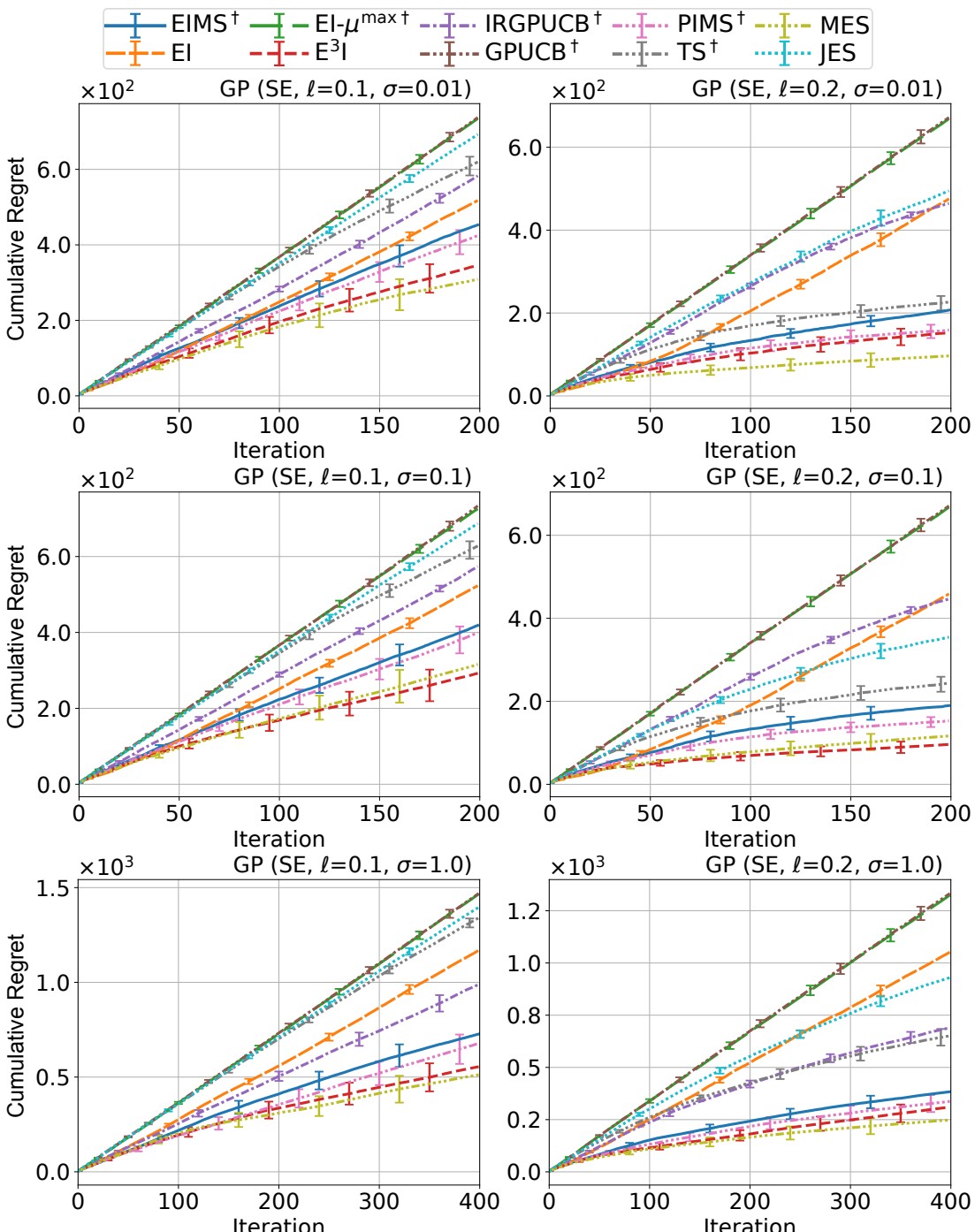

Figure 2: Results of cumulative regret for synthetic functions generated from a GP defined by the SE kernel. The top, middle, and bottom rows show the result for noise standard deviations $\sigma = 0.01, 0.1$, and 1, respectively. The left and right columns show the result for length scales of the kernel $\ell = 0.1$ and $\ell = 0.2$, respectively. Daggers in the legend indicate that the BO method was performed with theoretical settings, for example, regarding the hyperparameters.

MC sample (Takeno et al., 2024). Furthermore, GP-EIMS is the same as E$^3$I using one MC sample. Showing whether MES and E$^3$I have regret guarantees is important future work.

GP-EI shows superior performance for the simple regret, although its regret analysis for $\sigma > 0$ has not been shown. However, in the experiments for the cumulative regret, GP-EI clearly deteriorates compared with $E^3I$, MES, JES, GP-PIMS, and GP-EIMS, particularly for the case of $\ell = 0.2$. Since the noisy best observation $y_t^{\max}$ tends to be large with larger noise $\epsilon$, we can expect that GP-EI tends to lean more towards exploration. Since uncertainty sampling that maximizes $\sigma_{t-1}(\boldsymbol{x})$ achieves the simple regret upper bound that converges to zero (Vakili et al., 2021a), it is understandable that GP-EI shows superior performance for the simple regret[2]. On the other hand, we conjecture that the deterioration of GP-EI for the cumulative regret is due to over-exploration. Furthermore, we consider that since the objective function with $\ell = 0.1$ has many local optima, the aggressive exploration by larger $y_t^{\max}$ results in relatively better performance compared with that of $\ell = 0.2$.

GP-EIMS and GP-PIMS show superior performance consistently for both simple and cumulative regret, particularly compared with other theoretically guaranteed methods. Both GP-EIMS and GP-PIMS are based on the posterior sample of $g_t^*$ and show similar results in these experiments, although GP-EIMS chooses a larger $\sigma_{t-1}(\boldsymbol{x}_t)$ compared with GP-PIMS, due to the formulation of the AF.

We performed the experiments for the Matérn kernels $k_{\mathrm{Mat}}(\boldsymbol{x}, \boldsymbol{x}') = \frac{2^{1-\nu}}{\Gamma(\nu)} \left( \sqrt{2\nu} \frac{\|\boldsymbol{x} - \boldsymbol{x}'\|}{\ell} \right)^\nu K_\nu \left( \sqrt{2\nu} \frac{\|\boldsymbol{x} - \boldsymbol{x}'\|}{\ell} \right)$, where $\Gamma(\cdot)$ and $K_\nu(\cdot)$ are the gamma function and the modified Bessel function, and several benchmark functions from `https://www.sfu.ca/~ssurjano/optimization.html`. These experiments are shown in Appendix E. In the experiments for the Matérn kernels, we observed a similar tendency to the experiments for the SE kernel. In the experiments for the benchmark functions, we confirmed that all BO methods perform well in most cases since we employed heuristic parameter settings for GP-UCB, IRGP-UCB, and GP-EI-$\mu^{\max}$. On the other hand, GP-TS, GP-PIMS, and GP-EIMS show good performance without heuristic tuning, and GP-PIMS and GP-EIMS exhibit superior or comparable performance to GP-TS.

## 6 Conclusion

The regret analyses of GP-EI-based algorithms are fewer than those of theoretically established algorithms, such as GP-UCB and GP-TS, despite the fact that GP-EI-based algorithms have been utilized in various applications. Existing analysis by Wang & de Freitas (2014) requires increasing the posterior variance in the actual algorithm, which can deteriorate the optimization performance. Therefore, we analyzed the posterior sampling-based GP-EI algorithm called GP-EIMS, which does not require rescaling the posterior variance. We derived the sublinear BCR upper bounds of the GP-EIMS algorithm in the Bayesian setting. We demonstrated that GP-EIMS achieves at least comparable performance compared with other baseline methods, especially theoretically guaranteed EI-based methods, without sacrificing the theoretical guarantee.

**Limitation and Future Work:** Several future directions for this study exist. First, the posterior sampling for continuous $\mathcal{X}$ or discrete $\mathcal{X}$ with huge $|\mathcal{X}|$ requires some approximation, such as random Fourier features, in practice. Theoretical analysis incorporating such approximations (Mutny & Krause, 2018) is interesting. Second, since we show only the BCR upper bounds, whether GP-EIMS (and GP-PIMS) achieves sublinear high-probability cumulative regret bounds is of interest. Third, recent studies (Scarlett, 2018; Iwazaki, 2025) imply that the standard GP-UCB achieves a near-optimal regret upper bound with high probability. Showing whether other BO methods, including GP-EIMS, achieve a near-optimal regret upper bound is vital, as discussed in Section 4 in (Iwazaki, 2025). Lastly, our analysis for GP-EIMS concentrates on the Bayesian setting. As with the analysis of GP-TS in the frequentist setting (Chowdhury & Gopalan, 2017), theoretical analysis in the frequentist setting is also intriguing.

## Broader Impact Statement

Our work focuses on theoretical aspects of machine learning methods, and we do not anticipate any negative societal impact.

---

[2] Although Vakili et al. (2021b) analyze in the frequentist setting, their analysis immediately suggests a simple regret upper bound in the Bayesian setting.

**Acknowledgments**

This work was supported by JST ACT-X Grant Number (JPMJAX23CD and JPMJAX24C3), JST PRESTO Grant Number (JPMJPR24J6), JST CREST (JPMJCR21D3 and JPMJCR22N2), JST Moonshot R&D (JPMJMS2033-05), MEXT Program: Data Creation and Utilization Type Material Research and Development Project (Grant No. JPMXP1122712807), JSPS KAKENHI Grant Number (JP21H03498, JP23K19967, JP23K16943, JP24K20847, and JP25K03182), and RIKEN Center for Advanced Intelligence Project.

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

## A    Technical Issue in (Nguyen et al., 2017)

A counter-example against Lemma 7 in (Nguyen et al., 2017) can be made easily. Let us consider the BO algorithm that always evaluates the same point; that is, $\exists \boldsymbol{x} \in \mathcal{X}, \forall t \in [T], \boldsymbol{x}_t = \boldsymbol{x}$. Then, choosing an input $\tilde{\boldsymbol{x}}$ such that $k(\tilde{\boldsymbol{x}}, \tilde{\boldsymbol{x}}) - k(\boldsymbol{x}, \tilde{\boldsymbol{x}})^2 / k(\boldsymbol{x}, \boldsymbol{x}) \geq C$ with some strictly positive constant $C$, we can confirm $\sigma_t^2(\tilde{\boldsymbol{x}}) \geq k(\tilde{\boldsymbol{x}}, \tilde{\boldsymbol{x}}) - k(\boldsymbol{x}, \tilde{\boldsymbol{x}})^2 / k(\boldsymbol{x}, \boldsymbol{x}) \geq C > 0$ for all $t \in [T]$. Therefore, $\sum_{t=1}^T \sigma_t^2(\tilde{\boldsymbol{x}}) \geq CT = \Omega(T)$, which contradicts Lemma 7 in (Nguyen et al., 2017).

Note that if Lemma 7 in (Nguyen et al., 2017) is true, *any* BO algorithm achieves the converged simple regret upper bound in some settings. Suppose that $\sum_{t=1}^T \sigma_{t-1}^2(\boldsymbol{x}) = O(\gamma_T)$ for all $\boldsymbol{x} \in \mathcal{X}$ holds. Assume $\mathcal{X}$ is finite and $\beta_T$ is set so that for all $\boldsymbol{x} \in \mathcal{X}, |\mu_T(\boldsymbol{x}) - f(\boldsymbol{x})| \leq \beta_T^{1/2} \sigma_T(\boldsymbol{x})$ with high probability (for example,

as in Srinivas et al., 2010). Then, the following holds with high probability:

$$f(\boldsymbol{x}^*) - f(\hat{\boldsymbol{x}}_T) \leq \mu_T(\boldsymbol{x}^*) + \beta_T^{1/2}\sigma_T(\boldsymbol{x}^*) - \mu_T(\hat{\boldsymbol{x}}_T) + \beta_T^{1/2}\sigma_T(\hat{\boldsymbol{x}}_T) \tag{32}$$

$$\leq 2\beta_T^{1/2}\sigma_T(\tilde{\boldsymbol{x}}_T) \tag{33}$$

$$\leq \frac{1}{T}2\beta_T^{1/2}\sum_{t=1}^{T}\sigma_t(\tilde{\boldsymbol{x}}_T) \qquad (\text{because } \forall \boldsymbol{x} \in \mathcal{X}, \forall t \in [T], \sigma_t(\boldsymbol{x}) \geq \sigma_T(\boldsymbol{x})) \tag{34}$$

$$\overset{\text{False}}{=} O\left(\sqrt{\frac{\beta_T\gamma_T}{T}}\right), \tag{35}$$

where $\hat{\boldsymbol{x}}_T = \arg\max_{\boldsymbol{x}\in\mathcal{X}} \mu_T(\boldsymbol{x})$, $\tilde{\boldsymbol{x}}_T \coloneqq \arg\max_{\boldsymbol{x}\in\mathcal{X}} \sigma_T(\boldsymbol{x})$. This result suggests that (if the equality shown as $\overset{\text{False}}{=}$ is true) the simple regret upper bound converges to 0 when $\gamma_T\beta_T = o(T)$ regardless of the BO algorithm. For example, $\gamma_T\beta_T = o(T)$ holds for SE and Matérn kernels. This result is obviously strange.

## B  Regret Upper Bounds for GP-EI with Rescaling of Posterior Variance

This section shows the regret upper bounds for GP-EI with rescaling of the posterior variance in the Bayesian setting. The derivation of the regret upper bounds comes mainly from (Wang & de Freitas, 2014) and can be applied to the frequentist setting in almost the same way. For more details on the frequentist setting, we discuss in Appendix B.4. We denote the cumulative regret $R_T \coloneqq \sum_{t=1}^{T} f(\boldsymbol{x}^*) - f(\boldsymbol{x}_t)$.

### B.1  Auxiliary Lemmas

First, to generalize the analysis of (Wang & de Freitas, 2014), we provide the two lemmas modified slightly from (Wang & de Freitas, 2014).

**Lemma B.1** (Modified from Lemma 9 of (Wang & de Freitas, 2014))**.** *Pick $\boldsymbol{x} \in \mathcal{X}$ and $t \in [T]$. Suppose that $a \in \mathbb{R}$ satisfies the following inequality holds with some $\beta_t^{1/2} > 0$:*

$$|a - \mu_{t-1}(\boldsymbol{x})| \leq \beta_t^{1/2}\sigma_{t-1}(\boldsymbol{x}). \tag{36}$$

*Then, the following holds with $b \in \mathbb{R}$ and $\nu > 0$:*

$$\max\left\{(a-b)_+ - \beta_t^{1/2}\sigma_{t-1}(\boldsymbol{x}), \frac{\tau(-\beta_t^{1/2}/\nu^{1/2})}{\tau(\beta_t^{1/2}/\nu^{1/2})}(a-b)_+\right\} \tag{37}$$

$$\leq \nu^{1/2}\sigma_{t-1}(\boldsymbol{x})\tau\left(\frac{\mu_{t-1}(\boldsymbol{x})-b}{\nu^{1/2}\sigma_{t-1}(\boldsymbol{x})}\right) = \text{EI}\big(\mu_{t-1}(\boldsymbol{x}), \nu^{1/2}\sigma_{t-1}(\boldsymbol{x}), b\big) \tag{38}$$

$$\leq (a-b)_+ + (\beta_t^{1/2} + \nu^{1/2})\sigma_{t-1}(\boldsymbol{x}), \tag{39}$$

*where $(\cdot)_+ \coloneqq \max\{0, \cdot\}$.*

*Proof.* If $\sigma_{t-1}(\boldsymbol{x}) = 0$, then $a = \mu_{t-1}(\boldsymbol{x})$ and $\text{EI}\big(\mu_{t-1}(\boldsymbol{x}), \nu^{1/2}\sigma_{t-1}(\boldsymbol{x}), b\big) = (a-b)_+$. Therefore, the inequalities are trivial (Note that $\tau(\cdot)$ is a monotonically increasing function). Thus, we assume that $\sigma_{t-1}(\boldsymbol{x}) > 0$ hereafter. Furthermore, define the variables $q$ and $u$ as follows:

$$q = \frac{a-b}{\sigma_{t-1}(\boldsymbol{x})}, u = \frac{\mu_{t-1}(\boldsymbol{x})-b}{\sigma_{t-1}(\boldsymbol{x})}. \tag{40}$$

First, we derive the upper bound. From the definition,

$$\text{EI}\big(\mu_{t-1}(\boldsymbol{x}), \nu^{1/2}\sigma_{t-1}(\boldsymbol{x}), b\big) = \nu^{1/2}\sigma_{t-1}(\boldsymbol{x})\tau\left(\frac{u}{\nu^{1/2}}\right). \tag{41}$$

From the assumption, $|u - q| \leq \beta_t^{1/2}$. Since $\tau(\cdot)$ is monotonically increasing and $\tau(c) \leq 1 + c$ for $c > 0$, we obtain

$$\text{EI}\big(\mu_{t-1}(\boldsymbol{x}), \nu^{1/2}\sigma_{t-1}(\boldsymbol{x}), b\big) \leq \nu^{1/2}\sigma_{t-1}(\boldsymbol{x})\tau\left(\frac{(q)_+ + \beta_t^{1/2}}{\nu^{1/2}}\right) \tag{42}$$

$$\leq \nu^{1/2}\sigma_{t-1}(\boldsymbol{x})\left(\frac{(q)_+ + \beta_t^{1/2}}{\nu^{1/2}} + 1\right) \tag{43}$$

$$\leq (a - b)_+ + (\beta_t^{1/2} + \nu^{1/2})\sigma_{t-1}(\boldsymbol{x}). \tag{44}$$

Next, we derive the lower bound. If $(a - b)_+ = 0$, then the lower bound is trivial since the GP-EI acquisition function is non-negative. Therefore, assume $(a - b)_+ > 0$. Since $\tau(\cdot)$ is monotonically increasing, we can see that

$$\text{EI}\big(\mu_{t-1}(\boldsymbol{x}), \nu^{1/2}\sigma_{t-1}(\boldsymbol{x}), b\big) \geq \nu^{1/2}\sigma_{t-1}(\boldsymbol{x})\tau\left(\frac{q - \beta_t^{1/2}}{\nu^{1/2}}\right) \qquad \text{(because } q - \beta_t^{1/2} \leq u\text{)} \tag{45}$$

$$\geq \nu^{1/2}\sigma_{t-1}(\boldsymbol{x})\left(\frac{q - \beta_t^{1/2}}{\nu^{1/2}}\right) \qquad \text{(because } \tau(c) = c + \tau(-c) \geq c\text{)} \tag{46}$$

$$= (a - b)_+ - \beta_t^{1/2}\sigma_{t-1}(\boldsymbol{x}). \tag{47}$$

Furthermore, since $\tau(\cdot)$ is monotonically increasing,

$$\text{EI}\big(\mu_{t-1}(\boldsymbol{x}), \nu^{1/2}\sigma_{t-1}(\boldsymbol{x}), b\big) = \nu^{1/2}\sigma_{t-1}(\boldsymbol{x})\tau\left(\frac{\mu_{t-1}(\boldsymbol{x}) - a + a - b}{\nu^{1/2}\sigma_{t-1}(\boldsymbol{x})}\right) \tag{48}$$

$$\geq \nu^{1/2}\sigma_{t-1}(\boldsymbol{x})\tau\left(\frac{\mu_{t-1}(\boldsymbol{x}) - a}{\nu^{1/2}\sigma_{t-1}(\boldsymbol{x})}\right) \qquad \text{(because } a - b > 0 \text{ from assumption)} \tag{49}$$

$$\geq \nu^{1/2}\sigma_{t-1}(\boldsymbol{x})\tau\left(\frac{-\beta_t^{1/2}}{\nu^{1/2}}\right). \qquad \text{(because } \mu_{t-1}(\boldsymbol{x}) - a > -\beta_t^{1/2}\sigma_{t-1}(\boldsymbol{x})\text{)} \tag{50}$$

Hence, we obtain $\frac{\text{EI}\big(\mu_{t-1}(\boldsymbol{x}), \nu^{1/2}\sigma_{t-1}(\boldsymbol{x}), b\big)}{\nu^{1/2}\tau\big(-\beta_t^{1/2}/\nu^{1/2}\big)} \geq \sigma_{t-1}(\boldsymbol{x})$. By substuting this inequality to $\text{EI}\big(\mu_{t-1}(\boldsymbol{x}), \nu^{1/2}\sigma_{t-1}(\boldsymbol{x}), b\big) \geq (a - b)_+ - \beta_t^{1/2}\sigma_{t-1}(\boldsymbol{x})$, we can see that

$$\text{EI}\big(\mu_{t-1}(\boldsymbol{x}), \nu^{1/2}\sigma_{t-1}(\boldsymbol{x}), b\big) \geq \frac{\nu^{1/2}\tau\big(-\beta_t^{1/2}/\nu^{1/2}\big)}{\beta_t^{1/2} + \nu^{1/2}\tau\big(-\beta_t^{1/2}/\nu^{1/2}\big)}(a - b)_+ \tag{51}$$

$$= \frac{\nu^{1/2}\tau\big(-\beta_t^{1/2}/\nu^{1/2}\big)}{\nu^{1/2}\tau\big(\beta_t^{1/2}/\nu^{1/2}\big)}(a - b)_+. \tag{52}$$

$\square$

**Lemma B.2** (Modified from Lemma 10 of (Wang & de Freitas, 2014))**.** *Assume the kernel function* $k(\boldsymbol{x}, \boldsymbol{x}') \leq 1$ *for all* $\boldsymbol{x}, \boldsymbol{x}' \in \mathcal{X}$*. Let* $\boldsymbol{x}_t = \arg\max_{\boldsymbol{x} \in \mathcal{X}} \text{EI}\big(\mu_{t-1}(\boldsymbol{x}), \nu^{1/2}\sigma_{t-1}(\boldsymbol{x}), b\big)$ *with* $\min_{\boldsymbol{x} \in \mathcal{X}} \mu_{t-1}(\boldsymbol{x}) \leq b \leq \max_{\boldsymbol{x} \in \mathcal{X}} \mu_{t-1}(\boldsymbol{x})$ *and some* $\nu > 0$*. Then, the following inequality holds:*

$$b - \mu_{t-1}(\boldsymbol{x}_t) \leq \sqrt{\log(t - 1 + \sigma^2) - \log(\sigma^2)}\nu^{1/2}\sigma_{t-1}(\boldsymbol{x}_t). \tag{53}$$

*Proof.* If $b \le \mu_{t-1}(\boldsymbol{x}_t)$, then the inequality is trivial. Thus, we assume $b > \mu_{t-1}(\boldsymbol{x}_t)$ hereafter. From the assumption of $b$, there exists $\boldsymbol{z} \in \mathcal{X}$ such that $\mu_{t-1}(\boldsymbol{z}) \ge b$. Therefore, we obtain

$$\nu^{1/2}\sigma_{t-1}(\boldsymbol{z})\tau(0) \le \nu^{1/2}\sigma_{t-1}(\boldsymbol{z})\tau\left(\frac{\mu_{t-1}(\boldsymbol{z}) - b}{\nu^{1/2}\sigma_{t-1}(\boldsymbol{z})}\right) \tag{54}$$

$$= \mathrm{EI}\big(\mu_{t-1}(\boldsymbol{z}), \nu^{1/2}\sigma_{t-1}(\boldsymbol{z}), b\big) \tag{55}$$

$$\le \mathrm{EI}\big(\mu_{t-1}(\boldsymbol{x}_t), \nu^{1/2}\sigma_{t-1}(\boldsymbol{x}_t), b\big) \tag{56}$$

$$= \nu^{1/2}\sigma_{t-1}(\boldsymbol{x}_t)\tau\left(\frac{\mu_{t-1}(\boldsymbol{x}_t) - b}{\nu^{1/2}\sigma_{t-1}(\boldsymbol{x}_t)}\right). \tag{57}$$

Then, since we assume $b > \mu_{t-1}(\boldsymbol{x}_t)$ and $\tau(0) = \frac{1}{\sqrt{2\pi}}$, we can arrange the inequality as follows:

$$\frac{\sigma_{t-1}(\boldsymbol{z})}{\sqrt{2\pi}} \le \sigma_{t-1}(\boldsymbol{x}_t)\phi\left(\frac{\mu_{t-1}(\boldsymbol{x}_t) - b}{\nu^{1/2}\sigma_{t-1}(\boldsymbol{x}_t)}\right) \qquad \text{(because } \tau(c) < \phi(c) \text{ for } c < 0) \tag{58}$$

$$= \frac{\sigma_{t-1}(\boldsymbol{x}_t)}{\sqrt{2\pi}}\exp\left(-\frac{1}{2}\left(\frac{\mu_{t-1}(\boldsymbol{x}_t) - b}{\nu^{1/2}\sigma_{t-1}(\boldsymbol{x}_t)}\right)^2\right). \tag{59}$$

Hence, by applying the logarithm, we can see that

$$|\mu_{t-1}(\boldsymbol{x}_t) - b| \le \sqrt{2\log\left(\frac{\sigma_{t-1}(\boldsymbol{x}_t)}{\sigma_{t-1}(\boldsymbol{z})}\right)}\nu^{1/2}\sigma_{t-1}(\boldsymbol{x}_t) \tag{60}$$

$$= \sqrt{\log\left(\frac{\sigma_{t-1}^2(\boldsymbol{x}_t)}{\sigma_{t-1}^2(\boldsymbol{z})}\right)}\nu^{1/2}\sigma_{t-1}(\boldsymbol{x}_t). \tag{61}$$

We know that, for all $\boldsymbol{x} \in \mathcal{X}$, the posterior variance is bounded as $\sigma^2/(\sigma^2 + t - 1) \le \sigma_{t-1}^2(\boldsymbol{x}) \le 1$ from Lemma D.1 and monotone decreasing property of the posterior variance. By substituting $\sigma_{t-1}^2(\boldsymbol{x}_t) \le 1$ and $\sigma^2/(\sigma^2 + t - 1) \le \sigma_{t-1}^2(\boldsymbol{z})$, we can obtain the desired result. □

## B.2 Regret Bounds for GP-EI with Maximum Posterior Mean

In this section, although (Wang & de Freitas, 2014) use the parameters $\{\nu_t\}_{t \ge 1}$ to scale the posterior variance, we use $\{\beta_t(\delta)\}_{t \ge 1}$ instead of $\{\nu_t\}_{t \ge 1}$ for simplicity. We believe that this modification does not affect the essential order of the resulting regret upper bound.

**Theorem B.3.** *Let $f \sim \mathcal{GP}(0, k)$, where the kernel function $k$ is normalized as $k(\boldsymbol{x}, \boldsymbol{x}) \le 1$. Pick $\delta \in (0, 1)$. Assume $\mathcal{X}$ is a finite set or Assumption 2.1 holds.*

*(i) When $\mathcal{X}$ is a finite set, set $\beta_t(\delta) = 2\log(|\mathcal{X}|t^2\pi^2/(6\delta))$.*

*(ii) When Assumption 2.1 holds, set $\beta_t(\delta) = 2\log(\pi^2 t^2/(3\delta)) + 2d\log\left(\lceil bdrt^2\sqrt{\log(4ad/\delta)}\rceil + 1\right)$.*

*If $\boldsymbol{x}_t = \arg\max_{\boldsymbol{x} \in \mathcal{X}} \mathrm{EI}\big(\mu_{t-1}(\boldsymbol{x}), \beta_t^{1/2}(\delta)\sigma_{t-1}(\boldsymbol{x}), \mu_{t-1}^{\max}\big)$, where $\mu_{t-1}^{\max} = \max_{\boldsymbol{x} \in \mathcal{X}} \mu_{t-1}(\boldsymbol{x})$, then the cumulative regret is bounded from above with probability at least $1 - \delta$ as follows:*

$$R_T = O\left(\sqrt{\beta_T(\delta)\gamma_T T \log T}\right). \tag{62}$$

*Proof.* First, we provide the proof regarding case (i). In this case, from Lemma 5.1 of (Srinivas et al., 2010), the following holds:

$$\Pr\left(\forall \boldsymbol{x} \in \mathcal{X}, \forall t \in [T], |f(\boldsymbol{x}) - \mu_{t-1}(\boldsymbol{x})| \le \beta_t^{1/2}(\delta)\sigma_{t-1}(\boldsymbol{x})\right) \ge 1 - \delta. \tag{63}$$

Therefore, we assume $|f(\boldsymbol{x}) - \mu_{t-1}(\boldsymbol{x})| \leq \beta_t^{1/2}(\delta)\sigma_{t-1}(\boldsymbol{x})$ holds for all $\boldsymbol{x} \in \mathcal{X}$ and $t \in [T]$ hereafter. We can see that the instantaneous regret $r_t$ can be bounded as follows:

$$r_t = f(\boldsymbol{x}^*) - f(\boldsymbol{x}_t) \tag{64}$$

$$= f(\boldsymbol{x}^*) - \mu_{t-1}^{\max} + \mu_{t-1}^{\max} - f(\boldsymbol{x}_t) \tag{65}$$

$$\leq I_t(\boldsymbol{x}^*) + \mu_{t-1}^{\max} - \mu_{t-1}(\boldsymbol{x}_t) + \beta_t^{1/2}(\delta)\sigma_{t-1}(\boldsymbol{x}_t), \tag{66}$$

where $I_t(\boldsymbol{x}) = \left(f(\boldsymbol{x}) - \mu_{t-1}^{\max}\right)_+$. Regarding the term $I_t(\boldsymbol{x}^*)$, using Lemma B.1 with $a = f(\boldsymbol{x}^*)$, $b = \mu_{t-1}^{\max}$, and $\nu = \beta_t(\delta)$, we see that

$$I_t(\boldsymbol{x}^*) \leq C_3 \mathrm{EI}\left(\mu_{t-1}(\boldsymbol{x}^*), \beta_t^{1/2}(\delta)\sigma_{t-1}(\boldsymbol{x}^*), \mu_{t-1}^{\max}\right) \tag{67}$$

$$\leq C_3 \mathrm{EI}\left(\mu_{t-1}(\boldsymbol{x}_t), \beta_t^{1/2}(\delta)\sigma_{t-1}(\boldsymbol{x}_t), \mu_{t-1}^{\max}\right), \qquad \text{(because } \boldsymbol{x}_t \text{ is the maximizer)} \tag{68}$$

where $C_3 = \frac{\tau(1)}{\tau(-1)}$. Moreover, by applying Lemma B.1 again with $a = \mu_{t-1}(\boldsymbol{x}_t)$, $b = \mu_{t-1}^{\max}$, and $\nu = \beta_t(\delta)$, we can obtain

$$I_t(\boldsymbol{x}^*) \leq C_3 \left(\left(\mu_{t-1}(\boldsymbol{x}_t) - \mu_{t-1}^{\max}\right)_+ + 2\beta_t^{1/2}(\delta)\sigma_{t-1}(\boldsymbol{x}_t)\right) \tag{69}$$

$$\leq 2C_3 \beta_t^{1/2}(\delta)\sigma_{t-1}(\boldsymbol{x}_t). \qquad \left(\text{because } \mu_{t-1}^{\max} > \mu_{t-1}(\boldsymbol{x}_t)\right). \tag{70}$$

Furthermore, the term $\mu_{t-1}^{\max} - \mu_{t-1}(\boldsymbol{x}_t)$ is bounded from above by Lemma B.2:

$$\mu_{t-1}^{\max} - \mu_{t-1}(\boldsymbol{x}_t) \leq \sqrt{\log(t-1+\sigma^2) - \log(\sigma^2)}\beta_t^{1/2}(\delta)\sigma_{t-1}(\boldsymbol{x}_t). \tag{71}$$

Hence, aggregating the results, we can obtain

$$r_t \leq \left(2C_3 + \sqrt{\log(t-1+\sigma^2) - \log(\sigma^2)} + 1\right)\beta_t^{1/2}(\delta)\sigma_{t-1}(\boldsymbol{x}_t). \tag{72}$$

Then, the cumulative regret can be bounded from above as follows:

$$R_T = \sum_{t=1}^{T} r_t \tag{73}$$

$$\leq \sum_{t=1}^{T} \left(2C_3 + \sqrt{\log(t-1+\sigma^2) - \log(\sigma^2)} + 1\right)\beta_t^{1/2}(\delta)\sigma_{t-1}(\boldsymbol{x}_t) \tag{74}$$

$$\leq \left(2C_3 + \sqrt{\log(T-1+\sigma^2) - \log(\sigma^2)} + 1\right)\sum_{t=1}^{T}\beta_t^{1/2}(\delta)\sigma_{t-1}(\boldsymbol{x}_t) \tag{75}$$

$$\leq \left(2C_3 + \sqrt{\log(T-1+\sigma^2) - \log(\sigma^2)} + 1\right)\sqrt{\sum_{t=1}^{T}\beta_t(\delta)\sum_{t=1}^{T}\sigma_{t-1}^2(\boldsymbol{x}_t)} \tag{76}$$

$$\leq \left(2C_3 + \sqrt{\log(T-1+\sigma^2) - \log(\sigma^2)} + 1\right)\sqrt{C_1 T \beta_T(\delta)\gamma_T}. \tag{77}$$

In the second last and last inequalities, we use Cauchy–Schwarz inequality and the inequality $\sum_{t=1}^{T}\sigma_{t-1}^2(\boldsymbol{x}_t) \leq C_1\gamma_T$ (Srinivas et al., 2010).

For the case (ii), we consider a finite set $\mathcal{X}_t$ with each dimension evenly divided by $\tau_t = \left\lceil bdrt^2\sqrt{\log(4ad/\delta)}\right\rceil$, such that $|\mathcal{X}_t| = \tau_t^d$. In addition, we denote $[\boldsymbol{x}]_t = \arg\min_{\boldsymbol{x}' \in \mathcal{X}_t}\|\boldsymbol{x} - \boldsymbol{x}'\|_1$. For this $\mathcal{X}_t$, using the union bound and Lemmas 5.5, 5.6, and 5.7 of (Srinivas et al., 2010), the following events simultaneously hold with probability at least $1 - \delta$,

$$\forall t \in [T], \sup_{\boldsymbol{x} \in \mathcal{X}} |f(\boldsymbol{x}) - f([\boldsymbol{x}]_t)| \leq \frac{1}{t^2}, \tag{78}$$

$$\forall t \in [T], \forall \boldsymbol{x} \in \mathcal{X}_t \cup \{\boldsymbol{x}_t\}, |f(\boldsymbol{x}) - \mu_{t-1}(\boldsymbol{x})| \leq \beta_t^{1/2}(\delta)\sigma_{t-1}(\boldsymbol{x}), \tag{79}$$

since $\beta_t(\delta) \geq 2 \log \left( \frac{(|\mathcal{X}_t|+1)\pi^2 t^2}{3\delta} \right)$. Note that $+1$ in $|\mathcal{X}_t| + 1$ is required to guarantee the confidence bound on $\boldsymbol{x}_t$. Therefore, we assume the above two events hold hereafter.

Since the above two events hold, the instantaneous regret can be bounded as

$$r_t = f(\boldsymbol{x}^*) - f(\boldsymbol{x}_t) \tag{80}$$

$$= f(\boldsymbol{x}^*) - f([\boldsymbol{x}^*]_t) + f([\boldsymbol{x}^*]_t) - \mu_{t-1}^{\max} + \mu_{t-1}^{\max} - f(\boldsymbol{x}_t) \tag{81}$$

$$\leq \frac{1}{t^2} + \left( f([\boldsymbol{x}^*]_t) - \mu_{t-1}^{\max} \right)_+ + \mu_{t-1}^{\max} - \mu_{t-1}(\boldsymbol{x}_t) + \beta_t^{1/2}(\delta)\sigma_{t-1}(\boldsymbol{x}_t). \tag{82}$$

As with the case (i), using Lemma B.1 at $[\boldsymbol{x}^*]_t$ with $a = f([\boldsymbol{x}^*]_t)$, $b = \mu_{t-1}^{\max}$, and $\nu = \beta_t(\delta)$, we see that

$$\left( f([\boldsymbol{x}^*]_t) - \mu_{t-1}^{\max} \right)_+ \leq C_3 \text{EI}\left( \mu_{t-1}([\boldsymbol{x}^*]_t), \beta_t^{1/2}(\delta)\sigma_{t-1}([\boldsymbol{x}^*]_t), \mu_{t-1}^{\max} \right) \tag{83}$$

$$\leq C_3 \text{EI}\left( \mu_{t-1}(\boldsymbol{x}_t), \beta_t^{1/2}(\delta)\sigma_{t-1}(\boldsymbol{x}_t), \mu_{t-1}^{\max} \right). \qquad \text{(because } \boldsymbol{x}_t \text{ is the maximizer)} \tag{84}$$

Moreover, using Lemma B.1 again at $\boldsymbol{x}_t$ with $a = \mu_{t-1}(\boldsymbol{x}_t)$, $b = \mu_{t-1}^{\max}$, and $\nu = \beta_t(\delta)$, we can obtain

$$f([\boldsymbol{x}^*]_t) - \mu_{t-1}^{\max} \leq C_3 \left( \left( \mu_{t-1}(\boldsymbol{x}_t) - \mu_{t-1}^{\max} \right)_+ + 2\beta_t^{1/2}(\delta)\sigma_{t-1}(\boldsymbol{x}_t) \right) \tag{85}$$

$$\leq 2C_3 \beta_t^{1/2}(\delta)\sigma_{t-1}(\boldsymbol{x}_t). \tag{86}$$

Futhermore, using Lemma B.2,

$$\mu_{t-1}^{\max} - \mu_{t-1}(\boldsymbol{x}_t) \leq \sqrt{\log(t - 1 + \sigma^2) - \log(\sigma^2)}\beta_t^{1/2}(\delta)\sigma_{t-1}(\boldsymbol{x}_t). \tag{87}$$

In addition, $\sum_{t=1}^{T} 1/t^2 \leq \pi^2/6$. Therefore, repeating the same proof as case (i), we obtain

$$R_T \leq \frac{\pi^2}{6} + \left( 2C_3 + \sqrt{\log(T - 1 + \sigma^2) - \log(\sigma^2)} + 1 \right) \sqrt{C_1 T \beta_T(\delta)\gamma_T}. \tag{88}$$

$$\square$$

**Theorem B.4.** *Let $f \sim \mathcal{GP}(0, k)$, where the kernel function $k$ is normalized as $k(\boldsymbol{x}, \boldsymbol{x}) \leq 1$. Assume $\mathcal{X}$ is a finite set or Assumption 2.1 holds.*

*(i) When $\mathcal{X}$ is a finite set, set $\beta_t = 2 \log \left( |\mathcal{X}| t^2 / \sqrt{2\pi} + 1 \right)$.*

*(ii) When Assumption 2.1 holds, set $\beta_t = 2d \log \left( \lceil bdr t^2 (\sqrt{\log(ad)} + \sqrt{\pi}/2) \rceil \right) + 2 \log(t^2 / \sqrt{2\pi} + 1)$.*

*If $\boldsymbol{x}_t = \arg\max_{\boldsymbol{x} \in \mathcal{X}} \text{EI}\left( \mu_{t-1}(\boldsymbol{x}), \beta_t^{1/2}\sigma_{t-1}(\boldsymbol{x}), \mu_{t-1}^{\max} \right)$, where $\mu_{t-1}^{\max} = \max_{\boldsymbol{x} \in \mathcal{X}} \mu_{t-1}(\boldsymbol{x})$, then the BCR is bounded from above as follows:*

$$\text{BCR}_T = O \left( \sqrt{\beta_T \gamma_T T \log T} \right). \tag{89}$$

*Proof.* First, we consider the case (i).

$$\text{BCR}_T = \sum_{t=1}^{T} \mathbb{E}\left[ f(\boldsymbol{x}^*) - f(\boldsymbol{x}_t) \right] \tag{90}$$

$$= \sum_{t=1}^{T} \mathbb{E}\left[ f(\boldsymbol{x}^*) - U_t(\boldsymbol{x}^*) + U_t(\boldsymbol{x}^*) - \mu_{t-1}^{\max} + \mu_{t-1}^{\max} - \mu_{t-1}(\boldsymbol{x}_t) \right], \tag{91}$$

$$\leq \sum_{t=1}^{T} \mathbb{E}\left[ f(\boldsymbol{x}^*) - U_t(\boldsymbol{x}^*) + \left( U_t(\boldsymbol{x}^*) - \mu_{t-1}^{\max} \right)_+ + \mu_{t-1}^{\max} - \mu_{t-1}(\boldsymbol{x}_t) \right] \tag{92}$$

$$= \sum_{t=1}^{T} \mathbb{E}\left[ f(\boldsymbol{x}^*) - U_t(\boldsymbol{x}^*) \right] + \sum_{t=1}^{T} \mathbb{E}\left[ \left( U_t(\boldsymbol{x}^*) - \mu_{t-1}^{\max} \right)_+ \right] + \sum_{t=1}^{T} \mathbb{E}\left[ \mu_{t-1}^{\max} - \mu_{t-1}(\boldsymbol{x}_t) \right], \tag{93}$$

where $U_t(\boldsymbol{x}) = \mu_{t-1}(\boldsymbol{x}) + \beta_t^{1/2}\sigma_{t-1}(\boldsymbol{x})$. Then, from the same proof as Theorem B.1 of (Takeno et al., 2023), $\sum_{t=1}^{T} \mathbb{E}\left[f(\boldsymbol{x}^*) - U_t(\boldsymbol{x}^*)\right] \leq \pi^2/6$. Note that $\beta_t > 0$ due to the ceil function.

Regarding the term $\left(U_t(\boldsymbol{x}^*) - \mu_{t-1}^{\max}\right)_+$, we can apply Lemma B.1 at $\boldsymbol{x}^*$ with $a = U_t(\boldsymbol{x}^*)$, $b = \mu_{t-1}^{\max}$, and $\nu = \beta_t$. Therefore,

$$(U_t(\boldsymbol{x}^*) - \mu_{t-1}(\boldsymbol{x}_t))_+ \leq C_3 \text{EI}\left(\mu_{t-1}(\boldsymbol{x}^*), \beta_t^{1/2}\sigma_{t-1}(\boldsymbol{x}^*), \mu_{t-1}^{\max}\right) \tag{94}$$

$$\leq C_3 \text{EI}\left(\mu_{t-1}(\boldsymbol{x}_t), \beta_t^{1/2}\sigma_{t-1}(\boldsymbol{x}_t), \mu_{t-1}^{\max}\right), \tag{95}$$

where $C_3 = \frac{\tau(1)}{\tau(-1)}$. Moreover, using Lemma B.1 at $\boldsymbol{x}_t$ with $a = \mu_{t-1}(\boldsymbol{x})$, $b = \mu_{t-1}^{\max}$, and $\nu = \beta_t$, we obtain

$$(U_t(\boldsymbol{x}^*) - \mu_{t-1}(\boldsymbol{x}_t))_+ \leq C_3\left(\left(\mu_{t-1}(\boldsymbol{x}_t) - \mu_{t-1}^{\max}\right)_+ + 2\beta_t^{1/2}\sigma_{t-1}(\boldsymbol{x}_t)\right) = 2C_3\beta_t^{1/2}\sigma_{t-1}(\boldsymbol{x}_t). \tag{96}$$

Furthermore, regarding $\mu_{t-1}^{\max} - \mu_{t-1}(\boldsymbol{x}_t)$, using Lemma B.2,

$$\mu_{t-1}^{\max} - \mu_{t-1}(\boldsymbol{x}_t) \leq \sqrt{\log(t - 1 + \sigma^2) - \log(\sigma^2)}\beta_t^{1/2}\sigma_{t-1}(\boldsymbol{x}_t). \tag{97}$$

Hence, the sum of these terms can be bounded from above as with the proofs of Theorem B.3:

$$\sum_{t=1}^{T} \mathbb{E}\left[\left(U_t(\boldsymbol{x}^*) - \mu_{t-1}^{\max}\right)_+\right] + \sum_{t=1}^{T} \mathbb{E}\left[\mu_{t-1}^{\max} - \mu_{t-1}(\boldsymbol{x}_t)\right] \leq \left(2C_3 + \sqrt{\log(T - 1 + \sigma^2) - \log(\sigma^2)} + 1\right)\sqrt{C_1 T \beta_T \gamma_T}. \tag{98}$$

Consequently, we can obtain the following upper bound:

$$\text{BCR}_T \leq \frac{\pi^2}{6} + \left(2C_3 + \sqrt{\log(T - 1 + \sigma^2) - \log(\sigma^2)} + 1\right)\sqrt{C_1 T \beta_T \gamma_T}. \tag{99}$$

Next, for the case (ii),

$$\text{BCR}_T = \sum_{t=1}^{T} \mathbb{E}\left[f(\boldsymbol{x}^*) - f(\boldsymbol{x}_t)\right] \tag{100}$$

$$= \sum_{t=1}^{T} \mathbb{E}\left[f(\boldsymbol{x}^*) - f([\boldsymbol{x}^*]_t) + f([\boldsymbol{x}^*]_t) - U_t([\boldsymbol{x}^*]_t) + U_t([\boldsymbol{x}^*]_t) - \mu_{t-1}^{\max} + \mu_{t-1}^{\max} - \mu_{t-1}(\boldsymbol{x}_t)\right], \tag{101}$$

$$\leq \sum_{t=1}^{T} \mathbb{E}\left[f(\boldsymbol{x}^*) - f([\boldsymbol{x}^*]_t)\right] + \sum_{t=1}^{T} \mathbb{E}\left[f([\boldsymbol{x}^*]_t) - U_t([\boldsymbol{x}^*]_t)\right] \tag{102}$$

$$+ \sum_{t=1}^{T} \mathbb{E}\left[\left(U_t([\boldsymbol{x}^*]_t) - \mu_{t-1}^{\max}\right)_+\right] + \sum_{t=1}^{T} \mathbb{E}\left[\mu_{t-1}^{\max} - \mu_{t-1}(\boldsymbol{x}_t)\right], \tag{103}$$

where $U_t(\boldsymbol{x}) = \mu_{t-1}(\boldsymbol{x}) + \beta_t^{1/2}\sigma_{t-1}(\boldsymbol{x})$.

We consider a finite set $\mathcal{X}_t$ with each dimension evenly divided by $\tau_t = \left\lceil bdrt^2\left(\sqrt{\log(ad)} + \sqrt{\pi}/2\right)\right\rceil$, such that $|\mathcal{X}_t| = \tau_t^d$. In addition, we denote $[\boldsymbol{x}]_t = \arg\min_{\boldsymbol{x}' \in \mathcal{X}_t} \|\boldsymbol{x} - \boldsymbol{x}'\|_1$. Then, as with the proof of Theorem B.1 of (Takeno et al., 2023), we can obtain

$$\sum_{t=1}^{T} \mathbb{E}\left[f(\boldsymbol{x}^*) - f([\boldsymbol{x}^*]_t)\right] \leq \frac{\pi^2}{6}, \tag{104}$$

$$\sum_{t=1}^{T} \mathbb{E}\left[f([\boldsymbol{x}^*]_t) - U_t([\boldsymbol{x}^*]_t)\right] \leq \frac{\pi^2}{6}. \tag{105}$$

As with the case (i), using Lemma B.1 at $[\boldsymbol{x}^*]_t$ with $a = U_t([\boldsymbol{x}^*]_t)$, $b = \mu_{t-1}^{\max}$, and $\nu = \beta_t$, we see that

$$\left(U_t([\boldsymbol{x}^*]_t) - \mu_{t-1}^{\max}\right)_+ \leq C_3 \mathrm{EI}\left(\mu_{t-1}([\boldsymbol{x}^*]_t), \beta_t^{1/2}\sigma_{t-1}([\boldsymbol{x}^*]_t), \mu_{t-1}^{\max}\right) \tag{106}$$

$$\leq C_3 \mathrm{EI}\left(\mu_{t-1}(\boldsymbol{x}_t), \beta_t^{1/2}\sigma_{t-1}(\boldsymbol{x}_t), \mu_{t-1}^{\max}\right). \qquad (\boldsymbol{x}_t \text{ is the maximizer}) \tag{107}$$

Moreover, using Lemma B.1 again at $\boldsymbol{x}_t$ with $a = \mu_{t-1}(\boldsymbol{x}_t)$, $b = \mu_{t-1}^{\max}$, and $\nu = \beta_t$, we can obtain

$$\left(U_t([\boldsymbol{x}^*]_t) - \mu_{t-1}^{\max}\right)_+ \leq C_3 \left(\left(\mu_{t-1}(\boldsymbol{x}_t) - \mu_{t-1}^{\max}\right)_+ + 2\beta_t^{1/2}\sigma_{t-1}(\boldsymbol{x}_t)\right) \tag{108}$$

$$\leq 2C_3 \beta_t^{1/2}\sigma_{t-1}(\boldsymbol{x}_t). \tag{109}$$

Futhermore, using Lemma B.2,

$$\mu_{t-1}^{\max} - \mu_{t-1}(\boldsymbol{x}_t) \leq \sqrt{\log(t - 1 + \sigma^2) - \log(\sigma^2)}\beta_t^{1/2}\sigma_{t-1}(\boldsymbol{x}_t). \tag{110}$$

Therefore, repeating the same proof as Theorem B.3, we obtain

$$\mathrm{BCR}_T \leq \frac{\pi^2}{3} + \left(2C_3 + \sqrt{\log(T - 1 + \sigma^2) - \log(\sigma^2)} + 1\right)\sqrt{C_1 T \beta_T \gamma_T}. \tag{111}$$

$\square$

## B.3 Regret Bounds for GP-EI with Maximum Posterior Mean among Evaluated Points

First, we show a variant of Theorem B.3, where the reference value is changed to $\tilde{\mu}_{t-1}^{\max} = \max_{i \in [t-1]} \mu_{t-1}(\boldsymbol{x}_i)$:

**Theorem B.5.** *Let $f \sim \mathcal{GP}(0, k)$, where the kernel function $k$ is normalized as $k(\boldsymbol{x}, \boldsymbol{x}) \leq 1$. Pick $\delta \in (0, 1)$. Assume $\mathcal{X}$ is a finite set or Assumption 2.1 holds.*

*(i) When $\mathcal{X}$ is a finite set, set $\beta_t(\delta) = 2\log(|\mathcal{X}|t^2\pi^2/(6\delta))$.*

*(ii) When Assumption 2.1 holds, set $\beta_t(\delta) = 2\log(\pi^2 t^2/(3\delta)) + 2d\log\left(\left\lceil bdrt^2\sqrt{\log(4ad/\delta)}\right\rceil + t\right).$*

*If $\boldsymbol{x}_t = \arg\max_{\boldsymbol{x} \in \mathcal{X}} \mathrm{EI}\left(\mu_{t-1}(\boldsymbol{x}), \beta_t^{1/2}(\delta)\sigma_{t-1}(\boldsymbol{x}), \tilde{\mu}_{t-1}^{\max}\right)$, where $\tilde{\mu}_{t-1}^{\max} = \max_{i \in [t-1]} \mu_{t-1}(\boldsymbol{x}_i)$ for $t \geq 1$ and $\tilde{\mu}_0^{\max} = 0$ for $t = 1$, then the cumulative regret is bounded from above with probability at least $1 - \delta$ as follows:*

$$R_T = O\left(\sqrt{\beta_T(\delta)\gamma_T T \log T}\right). \tag{112}$$

*Proof.* As with the proof of Theorem B.3, we can guarantee the following events with probability at least $1 - \delta$:

$$\forall \boldsymbol{x} \in \mathcal{X}, |f(\boldsymbol{x}) - \mu_{t-1}(\boldsymbol{x})| \leq \beta_t^{1/2}(\delta)\sigma_{t-1}(\boldsymbol{x}), \tag{113}$$

for the case (i), and

$$\forall t \in [T], \sup_{\boldsymbol{x} \in \mathcal{X}} |f(\boldsymbol{x}) - f([\boldsymbol{x}]_t)| \leq \frac{1}{t^2}, \tag{114}$$

$$\forall t \in [T], \forall \boldsymbol{x} \in \mathcal{X}_t \cup \{\boldsymbol{x}_i\}_{i=1}^t, |f(\boldsymbol{x}) - \mu_{t-1}(\boldsymbol{x})| \leq \beta_t^{1/2}(\delta)\sigma_{t-1}(\boldsymbol{x}), \tag{115}$$

$$\tag{116}$$

for the case (ii). Note that, in the case (ii), we additionally consider the bounds on $\{\boldsymbol{x}_i\}_{i=1}^t$, not only $\boldsymbol{x}_t$, using $\beta_t(\delta) = \log\left(\frac{(|\mathcal{X}_t| + t)t^2\pi^2}{3\delta}\right)$ compared with the proof of Theorem B.3. In the remainder of the proof, we assume the above events hold in each case.

By almost the same proof as Theorem B.3, in which we set $a = f(\boldsymbol{x}_t)$ for a second application of Lemma B.1, we can obtain

$$r_t \le C_3 \tilde{I}_t(\boldsymbol{x}_t) + \left( 2C_3 + \sqrt{\log(t - 1 + \sigma^2) - \log(\sigma^2)} + 1 \right) \beta_t^{1/2}(\delta) \sigma_{t-1}(\boldsymbol{x}_t), \qquad \text{for the case (i),} \quad (117)$$

$$r_t \le \frac{1}{t^2} + C_3 \tilde{I}_t(\boldsymbol{x}_t) + \left( 2C_3 + \sqrt{\log(t - 1 + \sigma^2) - \log(\sigma^2)} + 1 \right) \beta_t^{1/2}(\delta) \sigma_{t-1}(\boldsymbol{x}_t), \quad \text{for the case (ii),} \quad (118)$$

where $\tilde{I}_t(\boldsymbol{x}_t) = (f(\boldsymbol{x}_t) - \tilde{\mu}_{t-1}^{\max})_+$ and $C_3 = \frac{\tau(1)}{\tau(-1)}$. Thus, the cumulative regret is bounded from above as follows:

$$R_T \le C_3 \sum_{t=1}^{T} \tilde{I}_t(\boldsymbol{x}_t) + \left( 2C_3 + \sqrt{\log(T - 1 + \sigma^2) - \log(\sigma^2)} + 1 \right) \sqrt{C_1 T \beta_T(\delta) \gamma_T}, \qquad \text{for the case (i),}$$
$$(119)$$

$$R_T \le \frac{\pi^2}{6} + C_3 \sum_{t=1}^{T} \tilde{I}_t(\boldsymbol{x}_t) + \left( 2C_3 + \sqrt{\log(T - 1 + \sigma^2) - \log(\sigma^2)} + 1 \right) \sqrt{C_1 T \beta_T(\delta) \gamma_T}, \quad \text{for the case (ii).}$$
$$(120)$$

Next, we obtain the upper bound of $\sum_{t=1}^{T} \tilde{I}_t(\boldsymbol{x}_t)$ by the proof modified slightly from (Tran-The et al., 2022; Hu et al., 2025). First, we can see that

$$\tilde{I}_1(\boldsymbol{x}_1) \le (f(\boldsymbol{x}_1) - 0)_+ \le \beta_1^{1/2}(\delta), \tag{121}$$

where we use the inequality $|f(\boldsymbol{x}_1)| \le \beta_1^{1/2}(\delta)\sqrt{k(\boldsymbol{x}_1, \boldsymbol{x}_1)} \le \beta_1^{1/2}(\delta)$ for both cases (i) and (ii). Then, define an indices $\mathcal{M} = \{m(i) \mid \tilde{I}_{m(i)}(\boldsymbol{x}_{m(i)}) > 0, m(i) > 1\}$ with ascending order $m(1) < \cdots < m(M)$, where $M := |\mathcal{M}|$. In addition, let $m(0) = m(1) - 1 \ge 1$. If $M = 0$, we can easily show $\sum_{t=1}^{T} \tilde{I}_t(\boldsymbol{x}_t) = \tilde{I}_1(\boldsymbol{x}_1) \le \beta_1^{1/2}(\delta)$. On the other hand, for the case of $M \ge 1$, we can obtain

$$\sum_{t=2}^{T} \tilde{I}_t(\boldsymbol{x}_t) = \sum_{i=1}^{M} f(\boldsymbol{x}_{m(i)}) - \tilde{\mu}_{m(i)-1}^{\max} \tag{122}$$

$$\le \sum_{i=1}^{M} f(\boldsymbol{x}_{m(i)}) - \mu_{m(i)-1}(\boldsymbol{x}_{m(i-1)}) \qquad (\text{because Definition of } \tilde{\mu}_{t-1}^{\max}) \tag{123}$$

$$\le \sum_{i=1}^{M} f(\boldsymbol{x}_{m(i)}) - f(\boldsymbol{x}_{m(i-1)}) + \beta_{m(i)}^{1/2}(\delta) \sigma_{m(i)-1}(\boldsymbol{x}_{m(i-1)}) \tag{124}$$

$$\le f(\boldsymbol{x}_{m(M)}) - f(\boldsymbol{x}_{m(0)}) + \sum_{i=1}^{M} \beta_{m(i)}^{1/2}(\delta) \sigma_{m(i-1)-1}(\boldsymbol{x}_{m(i-1)}) \tag{125}$$

$$\le 2\beta_1^{1/2}(\delta) + 2 + \sum_{t=1}^{T} \beta_t^{1/2}(\delta) \sigma_{t-1}(\boldsymbol{x}_t) \tag{126}$$

$$\le 2\beta_1^{1/2}(\delta) + 2 + \sqrt{C_1 T \beta_T(\delta) \gamma_T}, \tag{127}$$

where we use

- in the second inequality it is guaranteed that $\forall t \in [T], \forall \boldsymbol{x} \in \mathcal{X}, |f(\boldsymbol{x}) - \mu_{t-1}(\boldsymbol{x})| \le \beta_t^{1/2}(\delta)$ for the case (i), and $\forall t \in [T], \forall \boldsymbol{x} \in \mathcal{X}_t \cup \{\boldsymbol{x}_i\}_{i=1}^{t}, |f(\boldsymbol{x}) - \mu_{t-1}(\boldsymbol{x})| \le \beta_t^{1/2}(\delta) \sigma_{t-1}(\boldsymbol{x})$ for the case (ii) from the assumption;

- in the third inequality, $\sigma_{m(i)-1}(\boldsymbol{x}_{m(i-1)}) \le \sigma_{m(i-1)-1}(\boldsymbol{x}_{m(i-1)})$;

- in the fourth inequality, for all $\boldsymbol{x} \in \mathcal{X}, |f(\boldsymbol{x})| \le \beta_1^{1/2}$ for the case (i), and $|f(\boldsymbol{x})| \le |f([\boldsymbol{x}]_1)| + |f(\boldsymbol{x}) - f([\boldsymbol{x}]_1)| \le \beta_1^{1/2} + 1$ for the case (ii) from the assumption.

Consequently, $\sum_{t=1}^{T} \tilde{I}_t(\boldsymbol{x}_t) \leq 3\beta_1^{1/2}(\delta)+2+\sqrt{C_1 T \beta_T(\delta)\gamma_T} = O(\sqrt{\beta_T(\delta)\gamma_T T})$, which concludes the proof. $\square$

**Theorem B.6.** *Let $f \sim \mathcal{GP}(0,k)$, where the kernel function $k$ is normalized as $k(\boldsymbol{x},\boldsymbol{x}) \leq 1$. Assume $\mathcal{X}$ is a finite set or Assumption 2.1 holds.*

  *(i) When $\mathcal{X}$ is a finite set, set $\beta_t = 2\log\big(|\mathcal{X}|t^2/\sqrt{2\pi} + 1\big)$.*

  *(ii) When Assumption 2.1 holds, set $\beta_t = 2d\log\big(\lceil bdrt^2(\sqrt{\log(ad)} + \sqrt{\pi}/2)\rceil\big) + 2\log(t^2/\sqrt{2\pi} + 1)$.*

*If $\boldsymbol{x}_t = \arg\max_{\boldsymbol{x}\in\mathcal{X}} \mathrm{EI}\big(\mu_{t-1}(\boldsymbol{x}), \beta_t^{1/2}\sigma_{t-1}(\boldsymbol{x}), \tilde{\mu}_{t-1}^{\max}\big)$, where $\tilde{\mu}_{t-1}^{\max} = \max_{i\in[t-1]} \mu_{t-1}(\boldsymbol{x}_i)$ for $t \geq 1$ and $\tilde{\mu}_0^{\max} = 0$ for $t = 1$, then the BCR is bounded from above as follows:*

$$\mathrm{BCR}_T = O\left(\sqrt{\beta_T \gamma_T T \log T}\right). \tag{128}$$

*Proof.* By the same proof as Theorem B.4, for both cases (i) and (ii), we can see that

$$\mathrm{BCR}_T = C_3 \sum_{t=1}^{T} \mathbb{E}\big[\tilde{I}_t(\boldsymbol{x}_t)\big] + O\left(\sqrt{\beta_T \gamma_T T \log T}\right). \tag{129}$$

where $\tilde{I}_t(\boldsymbol{x}) = \big(f(\boldsymbol{x}) - \tilde{\mu}_{t-1}^{\max}\big)_+$ and $C_3 = \tau(1)/\tau(-1)$. Therefore, we here consider the upper bound of $\sum_{t=1}^{T} \mathbb{E}\big[\tilde{I}_t(\boldsymbol{x}_t)\big]$.

First, we can obtain

$$\mathbb{E}\big[\tilde{I}_1(\boldsymbol{x}_1)\big] = \mathbb{E}\big[(f(\boldsymbol{x}_1))_+\big] = 2\phi(0)\sqrt{k(\boldsymbol{x}_1,\boldsymbol{x}_1)} \leq \sqrt{\frac{2}{\pi}}, \tag{130}$$

as $f(\boldsymbol{x}_1) \sim \mathcal{N}(0, k(\boldsymbol{x}_1,\boldsymbol{x}_1))$. Then, define an indices $\mathcal{M} = \{m(i) \mid \tilde{I}_{m(i)}(\boldsymbol{x}_{m(i)}) > 0, m(i) > 1\}$ with ascending order $m(1) < \cdots < m(M)$, where $M := |\mathcal{M}|$. In addition, let $m(0) = m(1) - 1 \geq 1$. If $M = 0$, we can easily show $\sum_{t=1}^{T} \tilde{I}_t(\boldsymbol{x}_t) = \tilde{I}_1(\boldsymbol{x}_1) \leq \sqrt{\frac{2}{\pi}}$. Thus, we consider the case of $M \geq 1$. For any $\mathcal{M}$ under $M \geq 1$, we can obtain

$$\sum_{t=2}^{T} \tilde{I}_t(\boldsymbol{x}_t) = \sum_{i=1}^{M} f(\boldsymbol{x}_{m(i)}) - \tilde{\mu}_{m(i)-1}^{\max} \tag{131}$$

$$\leq \sum_{i=1}^{M} f(\boldsymbol{x}_{m(i)}) - \mu_{m(i)-1}(\boldsymbol{x}_{m(i-1)}) \tag{132}$$

$$= \sum_{i=1}^{M} f(\boldsymbol{x}_{m(i)}) - f(\boldsymbol{x}_{m(i-1)}) + f(\boldsymbol{x}_{m(i-1)}) - U_{m(i)}(\boldsymbol{x}_{m(i-1)}) + \beta_{m(i)}^{1/2}\sigma_{m(i)-1}(\boldsymbol{x}_{m(i-1)}) \tag{133}$$

$$\leq f(\boldsymbol{x}_{m(M)}) - f(\boldsymbol{x}_{m(0)}) + \sum_{i=1}^{M} f(\boldsymbol{x}_{m(i-1)}) - U_{m(i)}(\boldsymbol{x}_{m(i-1)}) + \sum_{i=1}^{M} \beta_{m(i)}^{1/2}\sigma_{m(i-1)-1}(\boldsymbol{x}_{m(i-1)}) \tag{134}$$

$$\leq \max_{\boldsymbol{x}\in\mathcal{X}} f(\boldsymbol{x}) - \min_{\boldsymbol{x}\in\mathcal{X}} f(\boldsymbol{x}) + \sum_{i=1}^{M} \big(f(\boldsymbol{x}_{m(i-1)}) - U_{m(i)}(\boldsymbol{x}_{m(i-1)})\big)_+ + \sum_{t=1}^{T} \beta_t^{1/2}\sigma_{t-1}(\boldsymbol{x}_t) \tag{135}$$

$$\leq \max_{\boldsymbol{x}\in\mathcal{X}} f(\boldsymbol{x}) - \min_{\boldsymbol{x}\in\mathcal{X}} f(\boldsymbol{x}) + \sum_{t=2}^{T}\sum_{j=1}^{t-1} (f(\boldsymbol{x}_j) - U_t(\boldsymbol{x}_j))_+ + \sum_{t=1}^{T} \beta_t^{1/2}\sigma_{t-1}(\boldsymbol{x}_t), \tag{136}$$

where $U_t(\boldsymbol{x}) = \mu_{t-1}(\boldsymbol{x}) + \beta_t^{1/2}\sigma_{t-1}(\boldsymbol{x})$. In the above transformation;

  • in the first inequality, we use the definition of $\tilde{\mu}_{m(i)-1}^{\max}$;

- in the second inequality, we use $\sigma_{m(i)-1}(\boldsymbol{x}_{m(i-1)}) \leq \sigma_{m(i-1)-1}(\boldsymbol{x}_{m(i-1)})$.

From Lemma D.2,

$$\mathbb{E}\left[\max_{\boldsymbol{x}\in\mathcal{X}} f(\boldsymbol{x}) - \min_{\boldsymbol{x}\in\mathcal{X}} f(\boldsymbol{x})\right] \leq 2\sqrt{2\log(|\mathcal{X}|/2) + 2}, \qquad\qquad \text{for the case (i),} \quad (137)$$

$$\mathbb{E}\left[\max_{\boldsymbol{x}\in\mathcal{X}} f(\boldsymbol{x}) - \min_{\boldsymbol{x}\in\mathcal{X}} f(\boldsymbol{x})\right] \leq 2\sqrt{2d\log\left(\lceil bdr(\log(ad) + \sqrt{\pi}/2)\rceil\right) + 2 - 2\log 2} + 2, \quad \text{for the case (ii).} \quad (138)$$

In addition, as with the proof of Theorem B.1 of (Takeno et al., 2023), we can obtain

$$\mathbb{E}\left[\sum_{t=2}^{T}\sum_{j=1}^{t-1}(f(\boldsymbol{x}_j) - U_t(\boldsymbol{x}_j))_+\right] \leq \sum_{t=1}^{T}\sum_{j=1}^{t-1}\frac{1}{t^2} \tag{139}$$

$$\leq \sum_{t=1}^{T}\frac{1}{t} \tag{140}$$

$$\leq 1 + \log T. \tag{141}$$

Furthermore, we know that $\sum_{t=1}^{T}\beta_t^{1/2}\sigma_{t-1}(\boldsymbol{x}_t) \leq \sqrt{C_1\beta_T T\gamma_T}$. Consequently, we can observe $\sum_{t=1}^{T}\mathbb{E}\left[\tilde{I}_t(\boldsymbol{x}_t)\right] = O(\sqrt{C_1\beta_T T\gamma_T})$. $\qquad\square$

### B.4 Discussion on Modification to Frequentist Setting

We focus on the analysis in the Bayesian setting. However, our proof approach for finite input domains can apply to the frequentist setting since, in general, (i) $\sup_{\boldsymbol{x}\in\mathcal{X}}|f(\boldsymbol{x})| \leq B$ with some constant $B$ from the assumption and (ii) the confidence bounds for $f$ hold for all $\boldsymbol{x} \in \mathcal{X}$ even if $\mathcal{X}$ is an infinite input domain (for example, Srinivas et al., 2010; Chowdhury & Gopalan, 2017). Therefore, since $\beta_t$ is set as $O(\sqrt{\gamma_T})$ to make confidnce bounds in the frequntist setting, our proof suggests that GP-EI with $\tilde{\mu}_t^{1/2}$ achieves $O(\sqrt{\beta_T\gamma_T T\log T}) = O(\gamma_T\sqrt{T\log T})$ cumulative regret upper bound with high probability without any other modification to the GP-EI algorithm like (Hu et al., 2025). To our knowledge, this has not been revealed by the existing studies.

Note that, in the frequentist setting, an extension to the expected regret upper bound from the high-probability regret bound is trivial under some conditions, such as $\sup_{\boldsymbol{x}\in\mathcal{X}} f(\boldsymbol{x}) \leq B$ and $\inf_{\boldsymbol{x}\in\mathcal{X}} f(\boldsymbol{x}) \geq -B$, which is assumed in most existing studies (for example, Srinivas et al., 2010; Chowdhury & Gopalan, 2017). If we obtain the high probability regret upper bound as $R_T = O(g(T)\log(1/\delta))$ with some increasing sublinear function $g(T) = o(T/\log(T))$, by setting $\delta = 1/T$, we see that

$$\mathbb{E}[R_T] \leq g(T)\log(1/\delta) + \delta\sum_{t=1}^{T} f(\boldsymbol{x}^*) - f(\boldsymbol{x}_t) \leq g(T)\log(T) + 2B, \tag{142}$$

which is sublinear. Similar derivation is often performed in the literature of the bandit algorithms, for example, discussed in (Abbasi-yadkori et al., 2011).

## C Proofs for GP-EIMS

Here, we show the proofs of the main paper's theorems, lemmas, and corollaries. Our proof flow is summarized as follows. First, from Lemma 4.1, we can see that

$$\text{BCR}_T \leq \sqrt{\sum_{t=1}^{T}\mathbb{E}\left[\eta_t^2 \mathbb{1}\{\eta_t \geq 0\}\right]}\sqrt{C_1\gamma_T}.$$

Then, for the case of $|\mathcal{X}| < \infty$, we show the upper bound of $\mathbb{E}\left[\eta_t^2 \mathbb{1}\{\eta_t \geq 0\}\right] = O(\log t)$ in Corollary 4.5, which is shown by Lemmas 4.2 and 4.4. Therefore, combining Lemma 4.1 and Corollary 4.5, we can obtain the sublinear BCR upper bound of $O(\sqrt{T\gamma_T \log T})$ as in Theorem 4.6. For the case that Assumption 2.1 holds, by adapting the credible interval obtained in Lemma 4.7, a similar proof derives the BCR upper bound of $O(\sqrt{T\gamma_T \log T})$ as in Theorem 4.8.

### C.1 Proof of Lemma 4.1

**Lemma 4.1.** *Let*

$$\eta_t := \frac{g_t^* - \mu_{t-1}(\boldsymbol{x}_t)}{\sigma_{t-1}(\boldsymbol{x}_t)}. \tag{143}$$

*Then, the BCR incurred by GP-EIMS can be bounded from above as follows:*

$$\mathrm{BCR}_T \leq \sqrt{\mathbb{E}\left[\sum_{t=1}^T \eta_t^2 \mathbb{1}\{\eta_t \geq 0\}\right]}\sqrt{C_1 \gamma_T}, \tag{144}$$

*where $C_1 := 2/\log(1+\sigma^{-2})$ and the indicator function $\mathbb{1}\{\eta_t \geq 0\} = 1$ if $\eta_t \geq 0$, and $0$ otherwise.*

*Proof.* From the tower property and linearity of expectation, we obtain

$$\mathrm{BCR}_T = \sum_{t=1}^T \mathbb{E}\left[f(\boldsymbol{x}^*) - f(\boldsymbol{x}_t)\right] \tag{145}$$

$$= \sum_{t=1}^T \mathbb{E}\left[f(\boldsymbol{x}^*) - g_t^* + g_t^* - f(\boldsymbol{x}_t)\right] \tag{146}$$

$$= \sum_{t=1}^T \mathbb{E}_{\mathcal{D}_{t-1}}\left[\mathbb{E}_t\left[f(\boldsymbol{x}^*) - g_t^*\right] + \mathbb{E}_t\left[g_t^* - f(\boldsymbol{x}_t)\right]\right] \tag{147}$$

$$\overset{(a)}{=} \sum_{t=1}^T \mathbb{E}_{\mathcal{D}_{t-1}}\left[\mathbb{E}_t\left[g_t^* - f(\boldsymbol{x}_t)\right]\right] \tag{148}$$

$$\overset{(b)}{=} \sum_{t=1}^T \mathbb{E}\left[g_t^* - \mu_{t-1}(\boldsymbol{x}_t)\right]. \tag{149}$$

For the equality (a), we use the fact $\mathbb{E}_t[f(\boldsymbol{x}^*)] = \mathbb{E}_t[g_t^*]$ since $f(\boldsymbol{x}^*) \mid \mathcal{D}_{t-1}$ and $g_t^* \mid \mathcal{D}_{t-1}$ are identically distributed. For the equality (b), we use the tower property of expectation and $\mathbb{E}_t[f(\boldsymbol{x}_t)] = \mathbb{E}_t[\mu_{t-1}(\boldsymbol{x}_t)]$, which holds since $\boldsymbol{x}_t$ defined via $g_t$ and $f$ are independent. Then, we define the random variable $\eta_t$ as follows:

$$\eta_t := \frac{g_t^* - \mu_{t-1}(\boldsymbol{x}_t)}{\sigma_{t-1}(\boldsymbol{x}_t)}. \tag{150}$$

Therefore, since $\sigma_{t-1}(\boldsymbol{x}) > 0$ due to $\sigma^2 > 0$, we obtain

$$\mathrm{BCR}_T = \sum_{t=1}^T \mathbb{E}\left[\eta_t \sigma_{t-1}(\boldsymbol{x}_t)\right]. \tag{151}$$

Furthermore, from the linearity of expectation, we can see that

$$\text{BCR}_T = \mathbb{E}\left[\sum_{t=1}^{T}\eta_t\sigma_{t-1}(\boldsymbol{x}_t)\right] \tag{152}$$

$$\leq \mathbb{E}\left[\sum_{t=1}^{T}\eta_t\mathbb{1}\{\eta_t \geq 0\}\sigma_{t-1}(\boldsymbol{x}_t)\right] \qquad (\text{because } \sigma_{t-1}(\boldsymbol{x}_t) > 0) \tag{153}$$

$$\leq \mathbb{E}\left[\sqrt{\sum_{t=1}^{T}\eta_t^2\mathbb{1}\{\eta_t \geq 0\}\sum_{t=1}^{T}\sigma_{t-1}^2(\boldsymbol{x}_t)}\right] \qquad (\text{because of Cauchy–Schwarz inequality }) \tag{154}$$

$$\leq \mathbb{E}\left[\sqrt{\sum_{t=1}^{T}\eta_t^2\mathbb{1}\{\eta_t \geq 0\}}\right]\sqrt{C_1\gamma_T}, \tag{155}$$

$$\leq \sqrt{\mathbb{E}\left[\sum_{t=1}^{T}\eta_t^2\mathbb{1}\{\eta_t \geq 0\}\right]}\sqrt{C_1\gamma_T}, \qquad (\text{because of Jensen's inequality}) \tag{156}$$

where $C_1 := 2/\log(1 + \sigma^{-2})$ and the indicator function $\mathbb{1}\{\eta_t \geq 0\} = 1$ if $\eta_t \geq 0$, and 0 otherwise. For the third inequality, we use $\sum_{t=1}^{T}\sigma_{t-1}^2(\boldsymbol{x}_t) \leq C_1\gamma_T$ (Lemma 5.2 in Srinivas et al., 2010). $\qquad\square$

## C.2 Proof of Lemma 4.2

**Lemma 4.2.** *Fix $\beta > 0$ and $U \leq \max_{\boldsymbol{x}\in\mathcal{X}}\{\mu_{t-1}(\boldsymbol{x})+\beta^{1/2}\sigma_{t-1}(\boldsymbol{x})\}$. Suppose that $k(\boldsymbol{x},\boldsymbol{x}) \leq 1$ for all $\boldsymbol{x}\in\mathcal{X}$. Then, the following inequality holds:*

$$\frac{U - \mu_{t-1}(\boldsymbol{x}_t)}{\sigma_{t-1}(\boldsymbol{x}_t)} \leq \sqrt{\log\left(\frac{\sigma^2 + t - 1}{\sigma^2}\right) + \beta} + \sqrt{2\pi\beta}, \tag{157}$$

*where $\boldsymbol{x}_t = \arg\max_{\boldsymbol{x}\in\mathcal{X}}\text{EI}(\mu_{t-1}(\boldsymbol{x}),\sigma_{t-1}(\boldsymbol{x}),U)$.*

*Proof.* Let $\boldsymbol{z}\in\mathcal{X}$ be the input that satisfies $U \leq \mu_{t-1}(\boldsymbol{z}) + \beta^{1/2}\sigma_{t-1}(\boldsymbol{z})$, that is $\frac{U-\mu_{t-1}(\boldsymbol{z})}{\sigma_{t-1}(\boldsymbol{z})} \leq \beta^{1/2}$. If $U \leq \mu_{t-1}(\boldsymbol{x}_t)$, it is obvious due to $\frac{U-\mu_{t-1}(\boldsymbol{x}_t)}{\sigma_{t-1}(\boldsymbol{x}_t)} \leq 0$. Thus, assume $U > \mu_{t-1}(\boldsymbol{x}_t)$ hereafter. Then, from the assumption and the monotonic increasing property of $\tau(c) = c\Phi(c) + \phi(c)$,

$$\text{EI}(\mu_{t-1}(\boldsymbol{z}),\sigma_{t-1}(\boldsymbol{z}),U) = \sigma_t(\boldsymbol{z})\tau\left(\frac{\mu_{t-1}(\boldsymbol{z}) - U}{\sigma_{t-1}(\boldsymbol{z})}\right) \tag{158}$$

$$\geq \sigma_t(\boldsymbol{z})\tau(-\beta^{1/2}). \tag{159}$$

In addition, due to the definition of $\boldsymbol{x}_t$, we can see that

$$\text{EI}(\mu_{t-1}(\boldsymbol{z}),\sigma_{t-1}(\boldsymbol{z}),U) \leq \text{EI}(\mu_{t-1}(\boldsymbol{x}_t),\sigma_{t-1}(\boldsymbol{x}_t),U) \tag{160}$$

Therefore, we obtain that

$$\sigma_t(\boldsymbol{z})\tau(-\beta^{1/2}) \leq \text{EI}(\mu_{t-1}(\boldsymbol{x}_t),\sigma_{t-1}(\boldsymbol{x}_t),U) \tag{161}$$

$$= \sigma_{t-1}(\boldsymbol{x}_t)\tau\left(\frac{\mu_{t-1}(\boldsymbol{x}_t) - U}{\sigma_{t-1}(\boldsymbol{x}_t)}\right). \tag{162}$$

From the assumption $U > \mu_{t-1}(\boldsymbol{x}_t)$, we can see that $\frac{\mu_{t-1}(\boldsymbol{x}_t)-U}{\sigma_{t-1}(\boldsymbol{x}_t)} < 0$. Then, since $\tau(c) \leq \phi(c)$ if $c < 0$, we further obtain

$$\sigma_t(\boldsymbol{z})\tau(-\beta^{1/2}) \leq \sigma_{t-1}(\boldsymbol{x}_t)\phi\left(\frac{\mu_{t-1}(\boldsymbol{x}_t) - U}{\sigma_{t-1}(\boldsymbol{x}_t)}\right) \tag{163}$$

$$= \frac{\sigma_{t-1}(\boldsymbol{x}_t)}{\sqrt{2\pi}}\exp\left(-\frac{(\mu_{t-1}(\boldsymbol{x}_t) - U)^2}{2\sigma_{t-1}^2(\boldsymbol{x}_t)}\right). \tag{164}$$

Hence, by transforming and applying the logarithm and the square root, we can obtain that

$$|\mu_{t-1}(\boldsymbol{x}_t) - U| \le \sqrt{2\log\left(\frac{\sigma_{t-1}(\boldsymbol{x}_t)}{\sqrt{2\pi}\sigma_t(\boldsymbol{z})\tau(-\beta^{1/2})}\right)}\sigma_{t-1}(\boldsymbol{x}_t). \tag{165}$$

Due to the monotonicity of $\sigma_{t-1}(\boldsymbol{x}) \ge \sigma_t(\boldsymbol{x})$ and the assumption of $k(\boldsymbol{x},\boldsymbol{x}) \le 1$, we see that $\sigma_{t-1}(\boldsymbol{x}_t) \le 1$. In addition, from Lemma D.1, $\sigma_{t-1}^2(\boldsymbol{z}) \ge \frac{\sigma^2}{\sigma^2+t-1}$. Therefore,

$$|\mu_{t-1}(\boldsymbol{x}_t) - U| \le \left(\sqrt{\log\left(\frac{\sigma^2+t-1}{\sigma^2}\right) - 2\log\left(\tau(-\beta^{1/2})\right) - \log(2\pi)}\right)\sigma_{t-1}(\boldsymbol{x}_t). \tag{166}$$

Then, for $\tau(-\beta^{1/2})$, we can bound as follows:

$$\tau(-\beta^{1/2}) = -\beta^{1/2}\Phi(-\beta^{1/2}) + \phi(-\beta^{1/2}) \tag{167}$$

$$= -\beta^{1/2}\left(1 - \Phi(\beta^{1/2})\right) + \phi(\beta^{1/2}) \tag{168}$$

$$\ge -\left(1 - \exp\left(-\sqrt{\frac{\pi}{2}}\beta^{1/2}\right)\right)\phi(\beta^{1/2}) + \phi(\beta^{1/2}) \tag{169}$$

$$= \exp\left(-\sqrt{\frac{\pi}{2}}\beta^{1/2}\right)\phi(\beta^{1/2}), \tag{170}$$

where we use Lemma 4.3. Thus, we obtain

$$-2\log\left(\tau(-\beta^{1/2})\right) \le -2\left(-\sqrt{\frac{\pi}{2}}\beta^{1/2} - \log\sqrt{2\pi} - \frac{\beta}{2}\right) \tag{171}$$

$$= \sqrt{2\pi\beta} + \log(2\pi) + \beta. \tag{172}$$

By combining the results, we obtain the desired result. $\qquad\square$

### C.3  Proofs for Sec. 4.2.1

First, we show the proof of the following corollary:

**Corollary 4.5.** *Assume the same condition as in Lemma 4.4. Suppose that $k(\boldsymbol{x},\boldsymbol{x}) \le 1$ for all $\boldsymbol{x} \in \mathcal{X}$. Then, by running GP-EIMS, the following inequality holds with probability at least $1 - \delta$:*

$$\eta_t \le \sqrt{\log\left(\frac{\sigma^2+t-1}{\sigma^2}\right) + \beta(\delta) + \sqrt{2\pi\beta(\delta)}}, \tag{173}$$

*where $\beta(\delta) = 2\log(|\mathcal{X}|/(2\delta))$. Furthermore, we obtain*

$$\mathbb{E}\left[\eta_t^2\mathbb{1}\{\eta_t \ge 0\}\right] \le \log\left(\frac{\sigma^2+t-1}{\sigma^2}\right) + C_2 + \sqrt{2\pi C_2}, \tag{174}$$

*where $C_2 = 2 + 2\log(|\mathcal{X}|/2)$.*

*Proof.* From Lemma 4.4, defining $\boldsymbol{z}_t = \arg\max_{\boldsymbol{x}\in\mathcal{X}} g_t(\boldsymbol{x})$, we see that

$$\Pr\left(g_t^* \le \mu_{t-1}(\boldsymbol{z}_t) + \beta^{1/2}(\delta)\sigma_{t-1}(\boldsymbol{z}_t)\right) \ge 1 - \delta, \tag{175}$$

since $g_t^* = g_t(\boldsymbol{z}_t)$. Therefore, as a direct consequence of Lemma 4.2, we obtain that

$$\eta_t \le \sqrt{\log\left(\frac{\sigma^2+t-1}{\sigma^2}\right) + \beta(\delta) + \sqrt{2\pi\beta(\delta)}}, \tag{176}$$

which holds with probability at least $1 - \delta$. Furthermore, since the right-hand side is positive, the following also holds:

$$\Pr\left(\eta_t^2 \mathbb{1}\{\eta_t \geq 0\} \leq \log\left(\frac{\sigma^2 + t - 1}{\sigma^2}\right) + \beta(\delta) + \sqrt{2\pi\beta(\delta)}\right) \geq 1 - \delta \tag{177}$$

$$\Leftrightarrow F\left(\log\left(\frac{\sigma^2 + t - 1}{\sigma^2}\right) + \beta(\delta) + \sqrt{2\pi\beta(\delta)}\right) \geq 1 - \delta, \tag{178}$$

where $F(\cdot)$ is the cumulative distribution function of $\eta_t^2 \mathbb{1}\{\eta \geq 0\}$. By applying $F^{-1}$, which is the generalized inverse function of $F$ and monotonically increasing, we obtain that

$$\log\left(\frac{\sigma^2 + t - 1}{\sigma^2}\right) + \beta(\delta) + \sqrt{2\pi\beta(\delta)} \geq F^{-1}(1 - \delta). \tag{179}$$

By substituting $Z \sim \mathrm{Uni}(0,1)$ to $\delta$ and taking the expectation with respect to $Z$,

$$\mathbb{E}_Z\left[\log\left(\frac{\sigma^2 + t - 1}{\sigma^2}\right) + \beta(Z) + \sqrt{2\pi\beta(Z)}\right] \geq \mathbb{E}_Z\left[F^{-1}(1 - Z)\right] = \mathbb{E}_Z\left[F^{-1}(Z)\right] = \mathbb{E}\left[\eta_t^2 \mathbb{1}\{\eta_t \geq 0\}\right]. \tag{180}$$

In the last equality, we use the fact that $\eta_t^2 \mathbb{1}\{\eta_t \geq 0\}$ and $F^{-1}(Z)$ are identically distributed. Furthermore, the left-hand side can be transformed as

$$\mathbb{E}_Z\left[\log\left(\frac{\sigma^2 + t - 1}{\sigma^2}\right) + \beta(Z) + \sqrt{2\pi\beta(Z)}\right] = \log\left(\frac{\sigma^2 + t - 1}{\sigma^2}\right) + \mathbb{E}_Z[\beta(Z)] + \mathbb{E}_Z[\sqrt{2\pi\beta(Z)}] \tag{181}$$

$$\leq \log\left(\frac{\sigma^2 + t - 1}{\sigma^2}\right) + \mathbb{E}_Z[\beta(Z)] + \sqrt{2\pi\mathbb{E}_Z[\beta(Z)]}, \tag{182}$$

where the last inequality is obtained by Jensen's inequality. In addition, we can derive

$$\mathbb{E}_Z[\beta(Z)] = 2\log(|\mathcal{X}|/2) + \mathbb{E}_Z[2\log(1/Z)] = 2\log(|\mathcal{X}|/2) + 2, \tag{183}$$

since $\log(1/Z)$ follows the exponential distribution. Consequently, we obtain that

$$\mathbb{E}\left[\eta_t^2 \mathbb{1}\{\eta_t \geq 0\}\right] \leq \log\left(\frac{\sigma^2 + t - 1}{\sigma^2}\right) + C_2 + \sqrt{2\pi C_2}, \tag{184}$$

where $C_2 = \mathbb{E}_Z[\beta(Z)] = 2\log(|\mathcal{X}|/2) + 2$. $\qquad\square$

Then, we provide the detailed proof of Theorem 4.6:

**Theorem 4.6.** *Let $f \sim \mathcal{GP}(0, k)$, where the kernel function $k$ is normalized as $k(\boldsymbol{x}, \boldsymbol{x}) \leq 1$, and $\mathcal{X}$ be finite. Then, by running GP-EIMS, the BCR can be bounded from above as follows:*

$$\mathrm{BCR}_T \leq \sqrt{C_1 B_T T \gamma_T}, \tag{185}$$

*where $B_T = \log\left(\frac{\sigma^2 + T - 1}{\sigma^2}\right) + C_2 + \sqrt{2\pi C_2}$, $C_1 := 2/\log(1 + \sigma^{-2})$, and $C_2 = 2 + 2\log(|\mathcal{X}|/2)$.*

*Proof.* From Lemma 4.1 and Corollary 4.5, we see that

$$\mathrm{BCR}_T \leq \sqrt{\sum_{t=1}^{T} \mathbb{E}\left[\eta_t^2 \mathbb{1}\{\eta_t \geq 0\}\right]} \sqrt{C_1 \gamma_T} \tag{186}$$

$$\leq \sqrt{C_1 B_T T \gamma_T}. \tag{187}$$

$$\square$$

### C.4 Proofs for Sec. 4.2.2

First we show the proof of Lemma 4.7:

**Lemma 4.7.** *Suppose Assumption 2.1 holds. Pick $\delta \in (0,1)$ and $t \geq 1$. Then, the following holds:*

$$\Pr\nolimits_t \left( \forall \boldsymbol{x} \in \mathcal{X}, f(\boldsymbol{x}) \leq \mu_{t-1}([\boldsymbol{x}]_t) + 2\beta_t^{1/2}(\delta/2)\sigma_{t-1}([\boldsymbol{x}]_t) \right) \geq 1 - \delta \tag{188}$$

*where $\beta_t(\delta) = 2d\log(m_t) - 2\log(2\delta)$ and $m_t = \max\left\{2, \left\lceil bdr\sqrt{(\sigma^2 + t - 1)\log(2ad)/\sigma^2} \right\rceil \right\}$.*

*Proof.* Under Assumption 2.1, using the union bound with respect to $j \in [d]$, we see

$$\Pr\left( \max_{j \in [d]} L_j > L(\delta) \right) \leq \delta, \tag{189}$$

where $L(\delta) := b\sqrt{\log(ad/\delta)}$. Therefore, the following inequality holds with probability at least $1 - \delta$:

$$\Pr\left( \forall \boldsymbol{x} \in \mathcal{X}, f(\boldsymbol{x}) - f([\boldsymbol{x}]_t) \leq L(\delta)\|\boldsymbol{x} - [\boldsymbol{x}]_t\|_1 \right) \geq 1 - \delta. \tag{190}$$

Hence, by using the union bound, the following upper bound holds with probability at least $1 - \delta$:

$$f(\boldsymbol{x}) - \mu_{t-1}([\boldsymbol{x}]_t) = f(\boldsymbol{x}) - f([\boldsymbol{x}]_t) + f([\boldsymbol{x}]_t) - \mu_{t-1}([\boldsymbol{x}]_t) \tag{191}$$

$$\leq L_{\delta/2}\|\boldsymbol{x} - [\boldsymbol{x}]_t\|_1 + \beta_t^{1/2}(\delta/2)\sigma_{t-1}([\boldsymbol{x}]_t). \tag{192}$$

In addition, from the construction of $\mathcal{X}_t$,

$$\forall \boldsymbol{x} \in \mathcal{X}, \|\boldsymbol{x} - [\boldsymbol{x}]_t\|_1 \leq \frac{dr}{m_t}. \tag{193}$$

Therefore, we see that

$$L_{\delta/2}\|\boldsymbol{x} - [\boldsymbol{x}]_t\|_1 \leq \frac{L_{\delta/2}dr}{m_t} \tag{194}$$

$$= \frac{bdr\sqrt{\log(2ad/\delta)}}{m_t} \tag{195}$$

$$\leq \sqrt{\frac{\sigma^2}{\sigma^2 + t - 1}}\sqrt{1 - \frac{\log(\delta)}{\log(2ad)}} \quad \left(\text{because } m_t \geq bdr\sqrt{(\sigma^2 + t - 1)\log(2ad)/\sigma^2}\right) \tag{196}$$

$$\leq \sqrt{\frac{\sigma^2}{\sigma^2 + t - 1}}\sqrt{1 - 2\log(\delta)}. \quad (\text{because } \log(2ad) \geq \log 2 > 0.69) \tag{197}$$

On the other hand,

$$\beta_t^{1/2}(\delta/2) = \sqrt{2d\log(m_t) - 2\log(\delta)} \tag{198}$$

$$\geq \sqrt{2d\log(2) - 2\log(\delta)}. \quad (\text{because } m_t \geq 2) \tag{199}$$

Moreover, from Lemma D.1, $\sigma_{t-1}^2(x) \geq \sigma^2/(\sigma^2 + t - 1)$ for all $\boldsymbol{x} \in \mathcal{X}$. Therefore, we obtain that

$$L_{\delta/2}\|\boldsymbol{x} - [\boldsymbol{x}]_t\|_1 \leq \beta_t^{1/2}(\delta/2)\sigma_{t-1}([\boldsymbol{x}]_t). \tag{200}$$

Finally, we derive the desired result:

$$f(\boldsymbol{x}) - \mu_{t-1}([\boldsymbol{x}]_t) \leq 2\beta_t^{1/2}(\delta/2)\sigma_{t-1}([\boldsymbol{x}]_t). \tag{201}$$

$$\square$$

Next, we show the BCR upper bound for continuous input domains:

**Theorem 4.8.** *Suppose Assumption 2.1 holds and $f \sim \mathcal{GP}(0, k)$, where the kernel function $k$ is normalized as $k(\boldsymbol{x}, \boldsymbol{x}) \leq 1$. Then, by running GP-EIMS, the BCR can be bounded from above as follows:*

$$\mathrm{BCR}_T \leq \sqrt{C_1 B_T T \gamma_T}, \tag{202}$$

*where $C_1 \coloneqq 2/\log(1 + \sigma^{-2})$, $B_T = \log\left(\frac{\sigma^2 + T - 1}{\sigma^2}\right) + C_T + \sqrt{2\pi C_T}$, $C_T = 8(d\log(m_T) + 1)$, and $m_T = \max\left\{2, \lceil bdr\sqrt{(\sigma^2 + T - 1)\log(2ad)/\sigma^2} \rceil\right\}$.*

*Proof.* As with the proof of Corollary 4.5, using Lemmas 4.2 and 4.7, we can derive that

$$\mathbb{E}\left[\eta_t^2 \mathbb{1}\{\eta_t \geq 0\}\right] \leq \log\left(\frac{\sigma^2 + t - 1}{\sigma^2}\right) + C_t + \sqrt{2\pi C_t}, \tag{203}$$

where $C_t = 8(d\log(m_t) + 1)$. Then, as with the proof of Theorem 4.6, we can obtain that

$$\mathrm{BCR}_T \leq \sqrt{\sum_{t=1}^{T} \mathbb{E}\left[\eta_t^2 \mathbb{1}\{\eta_t > 0\}\right]} \sqrt{C_1 \gamma_T} \tag{204}$$

$$\leq \sqrt{C_1 B_T T \gamma_T}. \tag{205}$$

$\square$

## D   Auxiliary Lemmas

**Lemma D.1** (Lemma 4.2 in (Takeno et al., 2024))**.** *Let $k$ be a kernel s.t. $k(\boldsymbol{x}, \boldsymbol{x}) \leq 1$. Then, the posterior variance is lower bounded as,*

$$\sigma_t^2(\boldsymbol{x}) \geq \frac{\sigma^2}{\sigma^2 + t}, \tag{206}$$

*for all $\boldsymbol{x} \in \mathcal{X}$ and for all $t \geq 0$.*

**Lemma D.2.** *Let $f \sim \mathcal{GP}(0, k)$, where $k$ is a kernel s.t. $k(\boldsymbol{x}, \boldsymbol{x}) \leq 1$. If $\mathcal{X}$ is finite, $\mathbb{E}[\max_{\boldsymbol{x} \in \mathcal{X}} f(\boldsymbol{x})] = -\mathbb{E}[\min_{\boldsymbol{x} \in \mathcal{X}} f(\boldsymbol{x})] \leq \sqrt{2\log(|\mathcal{X}|/2) + 2}$. If Assumption 2.1 holds, $\mathbb{E}[\max_{\boldsymbol{x} \in \mathcal{X}} f(\boldsymbol{x})] = -\mathbb{E}[\min_{\boldsymbol{x} \in \mathcal{X}} f(\boldsymbol{x})] \leq \sqrt{2d\log(\lceil bdr(\log(ad) + \sqrt{\pi}/2)\rceil) + 2 - 2\log 2} + 1$.*

*Proof.* First, since $f$ follows a centered GP, $\mathbb{E}[\max_{\boldsymbol{x} \in \mathcal{X}} f(\boldsymbol{x})] = -\mathbb{E}[\min_{\boldsymbol{x} \in \mathcal{X}} f(\boldsymbol{x})]$. For a finite input domain $\mathcal{X}$, from Lemma 4.2 of (Takeno et al., 2023), we can obtain

$$\mathbb{E}\left[\max_{\boldsymbol{x} \in \mathcal{X}} f(\boldsymbol{x})\right] \leq \mathbb{E}[\zeta^{1/2}] \leq \sqrt{\mathbb{E}[\zeta]} = \sqrt{2\log(|\mathcal{X}|/2) + 2}, \tag{207}$$

where $\zeta = 2\log(|\mathcal{X}|/2) + 2Z$ and $Z \sim \mathrm{Exp}(\lambda = 1)$.

For an infinite input domain, we consider a finite set $\mathcal{X}_t$ with each dimension evenly divided by $\tau_t = \lceil bdr(\log(ad) + \sqrt{\pi}/2)\rceil$, such that $|\mathcal{X}_t| = \tau_t^d$. In addition, we denote $[\boldsymbol{x}]_t = \arg\min_{\boldsymbol{x}' \in \mathcal{X}_t} \|\boldsymbol{x} - \boldsymbol{x}'\|_1$. Then, by Lemma H.2 of (Takeno et al., 2023), we see that

$$\mathbb{E}\left[f(\boldsymbol{x}^*) - f([\boldsymbol{x}^*]_t)\right] \leq 1. \tag{208}$$

Furthermore, as with the case of a finite input domain,

$$\mathbb{E}\left[\max_{\boldsymbol{x} \in \mathcal{X}_t} f(\boldsymbol{x})\right] \leq \sqrt{2\log(|\mathcal{X}_t|/2) + 2} = \sqrt{2d\log(\lceil bdr(\log(ad) + \sqrt{\pi}/2)\rceil) + 2 - 2\log 2}, \tag{209}$$

which concludes the proof. $\square$

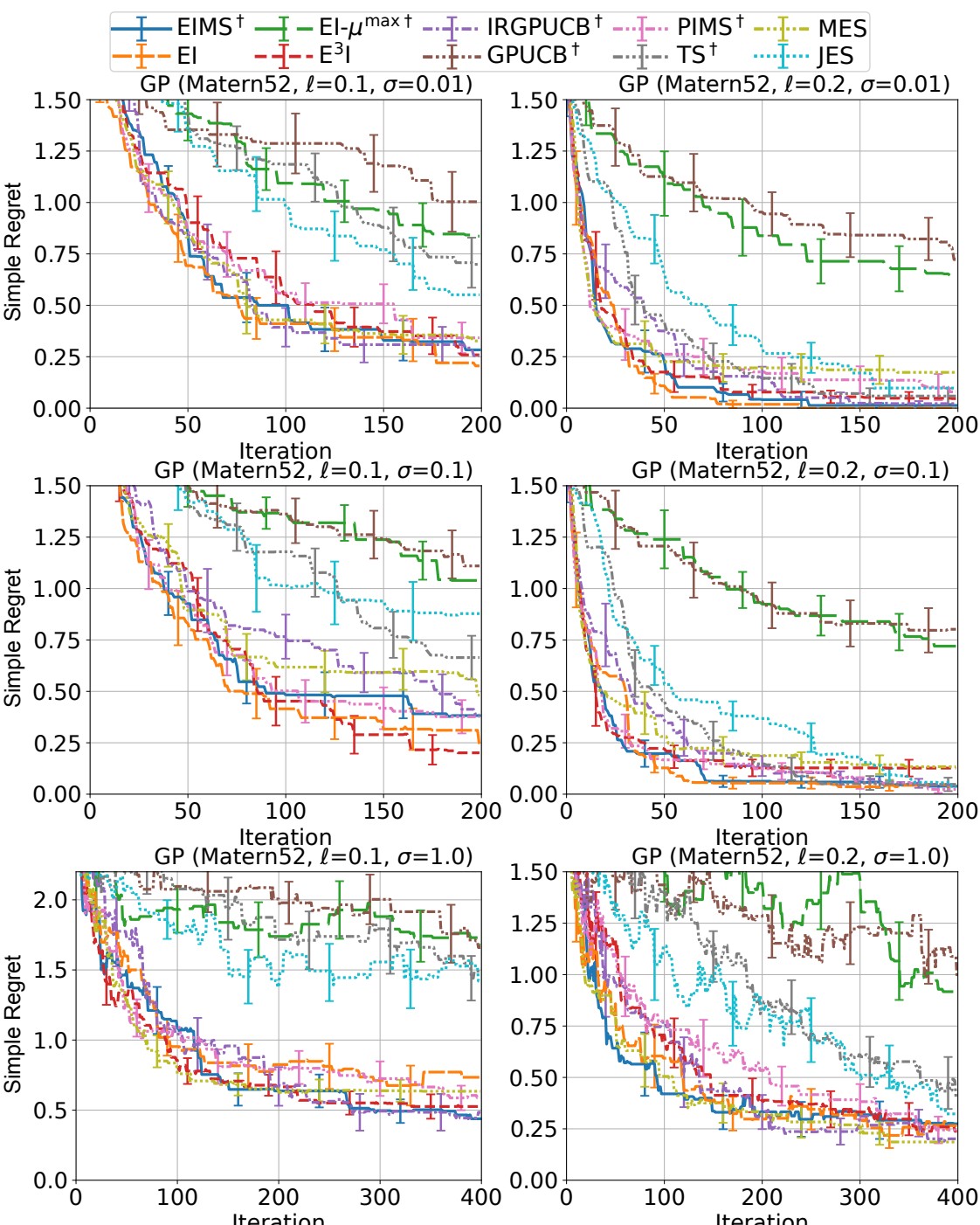

Figure 3: Results of simple regret for synthetic functions generated from a GP defined by the Matérn kernel with $\nu = 5/2$. The top, middle, and bottom rows show the result for noise standard deviations $\sigma = 0.01, 0.1$, and 1, respectively. The left and right columns show the result for length scales of the kernel $\ell = 0.1$ and $\ell = 0.2$, respectively. Daggers in the legend indicate that the BO method was performed with theoretical settings, for example, regarding the hyperparameters.

# E   Additional Experiments

## E.1   Synthetic Function Experiments with Matérn Kernel

Figures 3 and 4 show the simple regret and cumulative regret for the Matérn kernel with $\nu = 5/2$. The experimental settings are the same as those in the main paper, except for the kernel function. Also in these experiments, we can observe a similar tendency to the results of the SE kernel function.

## E.2   Benchmark Function Experiments

Figure 5 shows the results of the simple regret for several benchmark functions. We employed the SE kernel function with automatic relevance determination (Rasmussen & Williams, 2005). We performed hyperparameter selection for the GP model every five iterations. In these experiments, we did not add actual noise. Since $\arg\max_{\boldsymbol{x} \in \mathcal{X}} \mu_t(\boldsymbol{x})$ is unstable due to the hyperparameter selection, we define the simple regret using $\hat{\boldsymbol{x}}_t = \arg\max_{\boldsymbol{x} \in \mathcal{X}}\{\mu_t(\boldsymbol{x}) - 2\sigma_t(\boldsymbol{x})\}$. We employed the heuristic hyperparameters for the AFs of GP-UCB, IRGP-UCB, and GP-EI-$\mu^{\max}$ as $\beta_t = 0.2d\log(2t)$ as with (Kandasamy et al., 2016), $\nu_t = 0.2d\log(2t)$, and $\zeta_t = \max\{0.2d\log(2t) - 2, 0\} + Z_t$, where $Z_t \sim \text{Exp}(\lambda = 1/2)$. Therefore, GP-UCB, IRGP-UCB, and GP-EI-$\mu^{\max}$ are no longer theoretically guaranteed. Other settings are the same as those of the synthetic function experiments.

Except for the Ackley and Shekel functions, almost all BO methods perform well. For the Ackley function, GP-UCB, IRGP-UCB, and GP-EI show superior performance. This is because these BO methods are more exploitative than other BO methods in this experiment, and exploitative AFs work well for the Ackley function. However, for the Shekel function, more explorative GP-PIMS, GP-EIMS, and E$^3$I show superior performance. Therefore, we believe that these differences in performance come from the exploitation-exploration tradeoff. On the other hand, regarding the BO methods with the theoretical guarantee without heuristic tunings, we consider that GP-PIMS and GP-EIMS are superior or comparable to GP-TS.

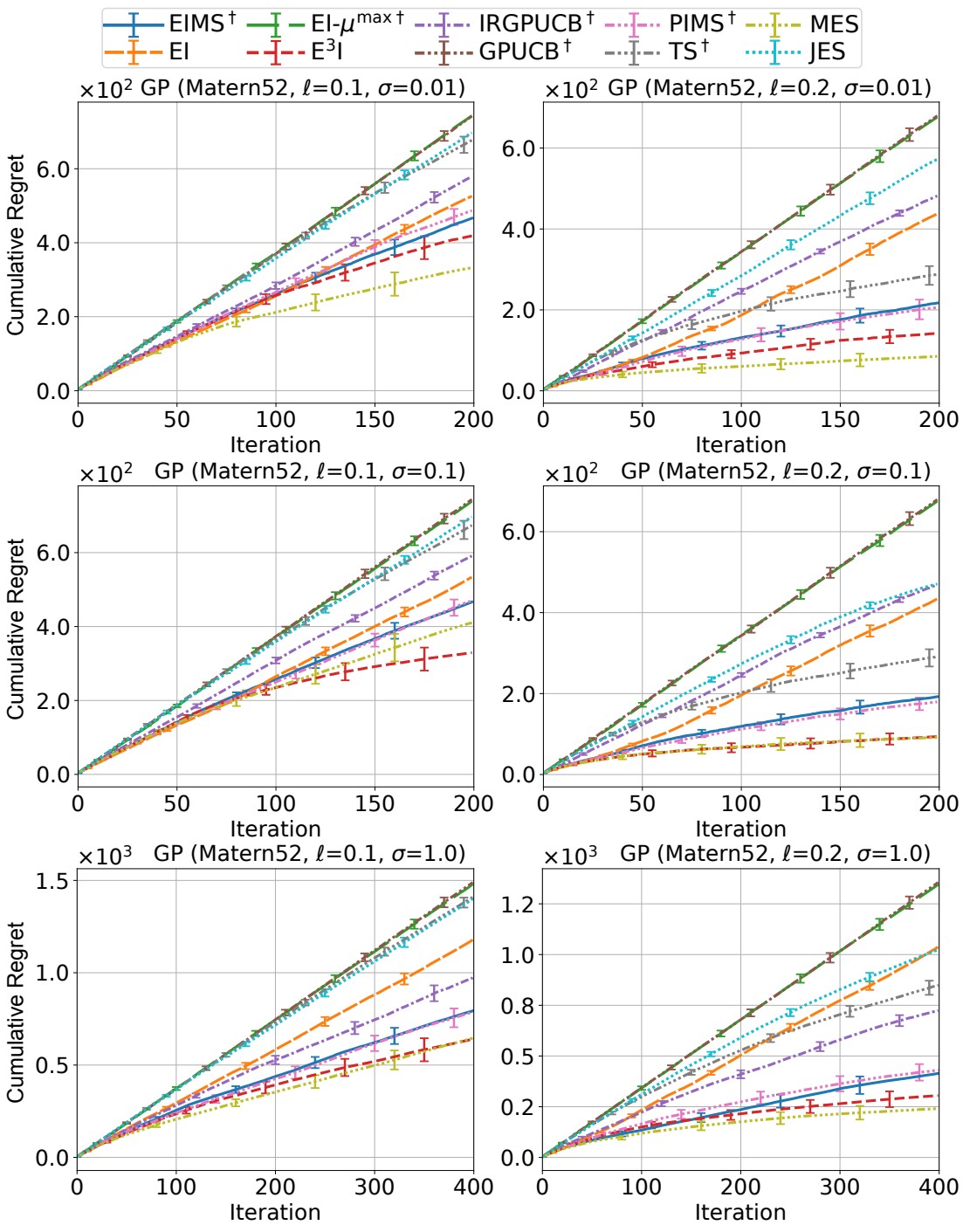

Figure 4: Results of cumulative regret for synthetic functions generated from a GP defined by the Matérn kernel with $\nu = 5/2$. The top, middle, and bottom rows show the result for noise standard deviations $\sigma = 0.01, 0.1$, and $1$, respectively. The left and right columns show the result for length scales of the kernel $\ell = 0.1$ and $\ell = 0.2$, respectively. Daggers in the legend indicate that the BO method was performed with theoretical settings, for example, regarding the hyperparameters.

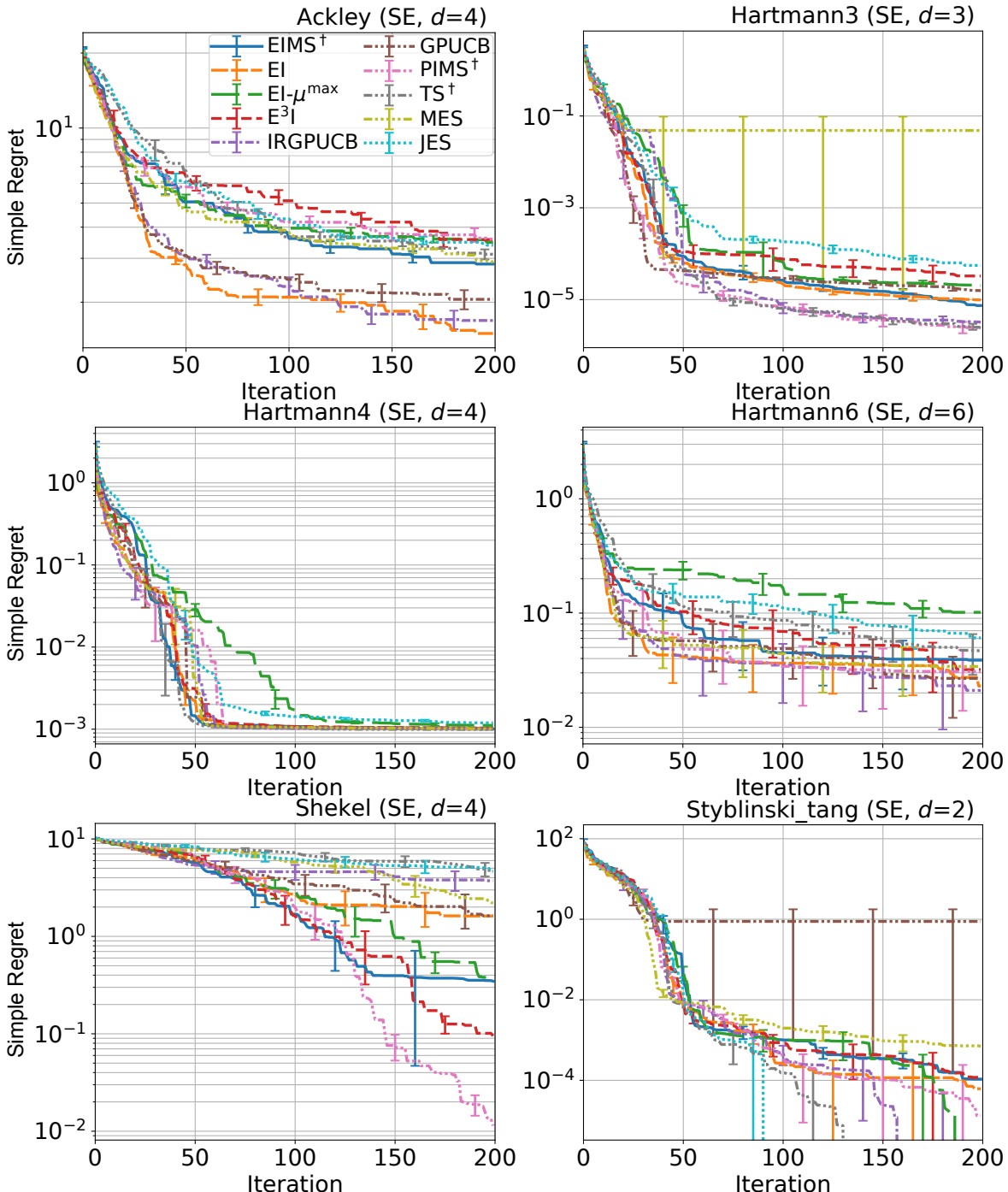

Figure 5: Results of simple regret for the benchmark functions. Daggers in the legend indicate that the BO method was performed with theoretical settings, for example, regarding the hyperparameters.

