# OpenReview forum: "Regret Analysis of Posterior Sampling-Based Expected Improvement for Bayesian Optimization"
_TMLR — Accepted by TMLR_

### Review · Reviewer_LpXR · 2025-07-31

**Summary Of Contributions:**

**Contributions / Paper Summary**

- **Sublinear Bayesian cumulative regret bounds for EIMS**. This paper analyzes the regret bounds of the Expected Improvement (EI) acquisition function when the improvement reference point is the maximum of a posterior sample path. The authors show that this posterior-sampling-based EI algorithm achieves sublinear cumulative Bayesian regret under standard assumptions.
  - Specifically, the authors adapt the general approach introduced by Takeno et al. (2024) to their proposed EI from the maximum of the sample path (EIMS). They further leverage results from Wang and de Freitas (2014), combined with a few clever observations, to derive the specific regret bounds.

- **Demonstrate the effectiveness of the proposal method with numerical experiments**. They also provide empirical evaluations on synthetic functions sampled from a Gaussian Process (GP) prior with known hyperparameters.

---

**Strengths**

**Theoretical Contribution**: The paper provides a novel regret analysis for a variant of EI that uses a posterior sample as the improvement reference. This analysis addresses an important gap in the literature and is relevant to the Bayesian optimization community.

**Contextualization with Prior Work**: The paper does a good job of situating its contribution within the context of recent related work. In particular, it discusses prior results and clarifies limitations and differences with previous analyses.

**Rigor**: The theoretical development appears to be technically sound, especially when the authors articulate misleading assumptions about past work.

---

**Weaknesses**

**Empirical Evaluation**: The experimental section is relatively weak and could improve significantly.

In my opinion, the cumulative regret plots, as currently presented, do not offer much insight. A summary table showing final simple regret values and statistical significance would be more informative and easier to interpret. Also, do some of the error bars have different colors?

**Test functions**: The experiments focus exclusively on synthetic functions sampled from the GP prior. While this setup is analytically convenient, it limits the practical relevance of the results. The paper would benefit from evaluating on standard benchmark test functions (such as those already listed in the appendix) to better demonstrate performance in practice, or at least a better proxy of that. For instance, selecting one function where an exploitative strategy performs well and another where exploration is crucial would replicate the current experimental conditions with the short and long length scales but in a more meaningful context.

Importantly, the authors emphasize throughout the paper that prior work relies on rescaling the posterior variance, which they argue hinders *practical optimization performance*. Given this motivation, expecting more than a validation experiment seems reasonable.

**Writing Quality**: While the logical flow of the paper is clear, the manuscript would benefit from a thorough language revision.

**Audience:**

Yes

**Audience Explanation:**

Yes, Expected Improvement is arguably the most widely used acquisition function in Bayesian optimization, so insights from its theoretical analysis are highly relevant to the audience.

**Broader Impact Concerns:**

No concerns here.

**Claims And Evidence:**

Yes

**Claims Explanation:**

- **Sublinear Bayesian cumulative regret bounds for EIMS**.  While I did not verify every technical detail, the theoretical analysis appears convincing. I cross-checked the results against two related papers and found the claims to be well-supported and credible.

- **Demonstrate the effectiveness of the proposal method with numerical experiments**. I find the experimental section unconvincing; however, I suspect that most readers of this paper may not place much weight on it.

**Requested Changes:**

**Minor Comments**

- **"Aims for optimization with fewer observations"**: Consider reprasing it to use fewest instead.

- **"BO sequentially queries a candidate determined by maximizing an acquisition function based on a Bayesian model"**: Consider being more precise — what is meant by “candidate” (e.g., input location)? or what is this "Bayesian model", of what?

- Consider replacing **"regret"** with **"regret bounds"**

- **"Performs querying an input"**: Suggest rephrasing to *“Performs queries on an input point/location”*

- The term **"twofold"** may not be appropriate in that context

- Consider using **"not advisable"** instead of **"not suggestive"**

- Adding a **y-axis label** would improve the interpretability of the plots.

---

> ### Author Response · Authors · 2025-08-22
>
> We appreciate the detailed and constructive comments.
> We have revised the paper with the updated parts highlighted in red font.
>
> >In my opinion, the cumulative regret plots, as currently presented, do not offer much insight.
>
> At first, the cumulative regret is a common performance measure in the literature of bandit problems.
> Also in the BO literature, the cumulative regret is one of the natural measures.
> For example, when considering the application of BO to A/B tests for web applications, intermediate evaluations, not just the final recommendation, are also important factors in evaluating performance.
> This is because user evaluations of poor web designs can become a burden.
>
> In our experiments, the cumulative regret plots demonstrate that GP-EI is inferior in terms of cumulative regret.
> We can see that the cumulative regret incurred by GP-EI seems to grow linearly.
> This is caused by over-exploration in later iterations, which occurs due to the dependence on $y^{\rm max}_t$ contaminated by noise.
> We still believe that cumulative regret plots are necessary to highlight the effectiveness of GP-EIMS compared to GP-EI.
> We have added a description clarifying this drawback of GP-EI compared with GP-EIMS.
>
> >A summary table showing final simple regret values and statistical significance would be more informative and easier to interpret.
>
> Although focusing only on the final simple regret can simplify the discussion as the reviewer suggested, we believe that evaluating the entire transition of the regret, not only the final simple regret, is essentially required to properly demonstrate the BO performance.
> In general, determining the optimal stopping time in BO is challenging, and thus, it can be inappropriate to evaluate performance based only on the final iteration $T$.
> For example, even if $\mathbb{E}[f(x^\star) - f(x_T)] \approx 0$, when $\mathbb{E}[f(x^\star) - f(x_{T-1})] \gg 0$, it cannot be concluded that the BO method is superior in the regret performance.
> Indeed, most existing studies [e.g., 1–7] discussed the overall transition of the regret rather than evaluating statistical significance solely on the final simple regret.
> We follow this practice in the BO literature.
>
> - [1] Kirthevasan Kandasamy, Gautam Dasarathy, Junier B. Oliva, Jeff Schneider, and Barnabas Póczos， Gaussian process bandit optimisation with multi-fidelity evaluations. In Advances in Neural Information Processing Systems 29, pp. 1000–1008. Curran Associates, Inc., 2016.
>
> - [2] Kirthevasan Kandasamy, Akshay Krishnamurthy, Jeff Schneider, and Barnabas Póczos. Parallelised Bayesian optimisation via Thompson sampling. In Proceedings of the 21st International Conference on Artificial Intelligence and Statistics, volume 84 of Proceedings of Machine Learning Research, pp. 133–142, 2018.
>
> - [3] Daniel Russo and Benjamin Van Roy. Learning to optimize via posterior sampling. Mathematics of Operations Research, 39(4):1221–1243, 2014
>
> - [4] Jasper Snoek, Hugo Larochelle, and Ryan P Adams. Practical Bayesian optimization of machine learning algorithms. In Advances in Neural Information Processing Systems 25, pp. 2951–2959. Curran Associates, Inc., 2012
>
> - [5] Niranjan Srinivas, Andreas Krause, Sham Kakade, and Matthias Seeger. Gaussian process optimization in the bandit setting: No regret and experimental design. In Proceedings of the 27th International Conference on Machine Learning, pp. 1015–1022. Omnipress, 2010.
>
> - [6] Sattar Vakili, Nacime Bouziani, Sepehr Jalali, Alberto Bernacchia, and Da-shan Shiu. Optimal order simple regret for Gaussian process bandits. In Advances in Neural Information Processing Systems, volume 34, pp. 21202–21215. Curran Associates, Inc., 2021
>
> - [7] Zi Wang and Stefanie Jegelka. Max-value entropy search for efficient Bayesian optimization. In Proceedings of the 34th International Conference on Machine Learning, volume 70 of Proceedings of Machine Learning Research, pp. 3627–3635, 2017
>
> >Also, do some of the error bars have different colors?
>
> We conjecture that the presence of several overlapping solid lines caused confusion.
> We have changed the line styles in the plots and enlarged the figures to improve visibility.

---

> ### Author Response · Authors · 2025-08-22
>
> >Test functions: The experiments focus exclusively on synthetic functions sampled from the GP prior. While this setup is analytically convenient, it limits the practical relevance of the results. The paper would benefit from evaluating on standard benchmark test functions (such as those already listed in the appendix) to better demonstrate performance in practice, or at least a better proxy of that. For instance, selecting one function where an exploitative strategy performs well and another where exploration is crucial would replicate the current experimental conditions with the short and long length scales but in a more meaningful context.
>
> >Given this motivation, expecting more than a validation experiment seems reasonable.
>
> We have performed the experiments for standard test functions in Appendix E.
> BO methods that we compared perform well in our experiments, where we employ widely used heuristics for selecting the hyperparameters of AFs.
> However, importantly, the BO methods that run with the theoretical settings are only GP-TS, GP-PIMS, and GP-EIMS since we employ the empirically recommended hyperparameters for other theoretically guaranteed methods, such as GP-UCB and EI-$\mu^{\rm max}$.
> We believe that our experiments demonstrate that GP-EIMS has superior or comparable performance to other BO methods without sacrificing the theoretical guarantee.
>
> >Importantly, the authors emphasize throughout the paper that prior work relies on rescaling the posterior variance, which they argue hinders practical optimization performance.
>
> We would like to claim that GP-EIMS achieves both sublinear BCR upper bounds and good optimization performance without any practical heuristics, especially compared with other theoretically guaranteed EI-based methods such as EI-$\mu^{\rm max}$.
> In the synthetic function experiments, EI-$\mu^{\rm max}$ clearly deteriorated.
> In the experiments for the benchmark functions, EI-$\mu^{\rm max}$ are no longer theoretically guaranteed, since they employed heuristic tuning of hyperparameters.
> We have revised the paper to clarify this claim.
>
> >Other requested changes
>
> We reflected on all of the reviewers' suggestions.

---

### Review · Reviewer_mJGG · 2025-07-31

**Summary Of Contributions:**

This paper proposes GP-EIMS, a new posterior sampling–based variant of Expected Improvement (EI), in which the *reference value* is set to the maximum of a single posterior sample path ($g_t^*$) at each iteration $t$. The key theoretical contribution lies in an elaboration of issues in contemporary regret-proof on EI acquisition function, and the proof of sublinear Bayesian cumulative regret (BCR) [1] for GP-EIMS under standard GP prior assumptions with stationary kernels (squared exponential, linear, Matérn families with sufficient smoothness). Empirically, the algorithm has been compared with different contemporary acquisition functions on synthetic functions ranging from $d=1\sim d=6$ and demonstrates interesting results among the acquisition functions with theoretical garantee. Since the main contribution is on theoretical analysis, a proof sketch summary is provided at the next block.

Strength:
- The paper provides a theoretical proof of Bayesian cumulative regret for both discrete and continuous domains (under strong assumptions), which is rigorous and plausible.
- It provides an extensive discussion of the limitations in contemporary regret analyses for the Expected Improvement (EI) acquisition function.
- The paper clearly articulates its contributions, assumptions, and limitations.

Weaknesses:

- Algorithmic:
    - The proposed method is somewhat incremental, as it can be viewed as a special degenerate case of the E3I framework.
    - The analysis focuses solely on Bayesian cumulative regret, which provides an average-case guarantee. Unlike methods such as GP-UCB, it does not establish high-probability regret bounds, limiting its robustness under adversarial scenarios.
- Theoretical:
    - In the continuous setting, the analysis assumes exact maximization of posterior samples. As acknowledged in the paper’s limitations, this requires a strong idealization that disregards approximation artifacts such as random Fourier features (RFF). Strictly speaking, the proof in the continuous domain is not fully complete without addressing such approximation errors.
- Empirical:
    - Experiments are limited to small to middle-scale, discrete search spaces, which restricts the demonstration of practical scalability.
    - Performance is comparable to or slightly worse than PIMS, casting doubt on the method’s empirical advantage. In particular, it is counterintuitive that the PI-based posterior sampling variant outperforms the EI-based one, despite the theoretical motivation. Additionally, stronger baselines such as Joint Entropy Search are not considered, further weakening the empirical positioning.

**Additional Comments:**

**Question**:
Since it is assumed that hyperparameters are known within this research, could the author provide some comments on how the regret bound degrades under GP model misspecification (e.g., mismatched kernel, unknown hyperparameters)?

**Audience:**

Yes

**Audience Explanation:**

This paper will be of interest to the Bayesian Optimization and GP bandit communities, as it provides new progress on the regret analysis of the Expected Improvement (EI) acquisition function. In particular, it offers a rigorous sublinear Bayesian cumulative regret bound for a posterior-sampling-based variant of EI (GP-EIMS), which has been historically difficult to analyze. Moreover, the work sheds light on overlooked issues in prior regret proofs for EI-type strategies, offering a clearer theoretical foundation for future research.

**Claims And Evidence:**

Yes

**Claims Explanation:**

The main claims in the submission are theoretical, focusing on bounding the Bayesian cumulative regret (BCR) for a posterior-sampling-based variant of the Expected Improvement (EI) acquisition function. A proof sketch, to the best of my understanding, is outlined below:

----------------
Proof Sketch
- **Bounding Regret via Normalized Deviation (Lemma 4.1)**:

    \begin{equation}
       BCR_T \leq
        \underbrace{\left( \mathbb{E} \left[  \sum_{t=1}^T \eta_t^2 \cdot \mathbb{I} (\eta_t \geq 0 )  \right] \right)^{1/2}}_{\text{Expected squared normalized deviations}}
        \cdot \sqrt{ C_1 \gamma_T },
    \end{equation}
 The problem reduces to bounding the accumulation of normalized deviations $\eta_t$.
- **Uniform Upper Bound on Normalized Deviation**:
By constructing a high-probability upper bound $U$ on the sample path maximum $ g_t^*$, the normalized deviation satisfies:

\begin{equation}
 \eta_t  \leq \frac{U - \mu_{t-1}(\mathbf x_t)}{\sigma_{t-1}(\mathbf x_t)} \leq \sqrt{\log\left( \frac{\sigma^2 + t - 1}{\sigma^2} \right)} + \beta + \sqrt{2\pi\beta}
\end{equation}

- **Bound the expectation term**:
$\mathbb{E} \left[ \eta_t^2 \cdot \mathbb{I}(\eta_t \geq 0) \right] \leq B_t$
and
$\mathbb{E} \left[ \sum_{t=1}^T \eta_t^2 \cdot \mathbb{I}(\eta_t \geq 0) \right] \leq \sum_{t=1}^T B_t = \mathcal{O}(T \log T)$
 and eventually upper bound on the accumulation of normalized deviations and hence BCR$_T$.

While I am not deeply familiar with the full BCR literature, I have examined the derivations in Appendix C and found them plausible and technically sound. The logical steps appear coherent and well-supported.

**Requested Changes:**

There are a few points that could be considered to strengthen the paper:
- **Comparison with $E^3I$**: one important aspect, since this-EI variant can be regarded as a degenrate version of $E^3I$, it is important to understand its empirical performance compared with it. It is not expected that EIMC is empirically better than $E^3I$, but this provide a better position of this paper in terms of its empirical performance.
- **Clarification of randomness in expectations**: The statement "the expectation is taken with all the randomness, that is, $f$, $\epsilon$, and the randomness of the algorithm'' should be clarified. Specifically, the ``randomness of the algorithm'' refers only to the sampling of $g_t$ from the posterior GP, i.e., $g_t \sim \mathcal{GP}(\mu_{t-1}, k_{t-1})$. Otherwise, this may be misleading.
- **Inaccurate citation**?: The citation ``Kandasamy et al., 2018'' appears to focus on Bayesian **simple regret** rather than Bayesian Cumulative Regret (BCR). It would be more accurate to either remove this citation or clarify the distinction.
-  **Justification for BCR focus**: For accessibility to a broader audience, it is recommended to include a sentence justifying the choice of BCR as the analysis objective, instead of the more commonly used high-probability cumulative regret bounds (e.g., those in GP-UCB)
-  **Improved proof accessibility**: While the paper is mostly clearly written, the regret proof in Appendix C remains difficult to follow without substantial background. i strongly recommend including a proof sketch or a high-level discussion at the beginning of the theoretical section. This would help guide readers by outlining the logical structure of the proof.

---

> ### Author Response · Authors · 2025-08-22
>
> We appreciate the detailed and constructive comments.
> We have revised the paper with the updated parts highlighted in red font.
>
> >Performance is comparable to or slightly worse than PIMS, casting doubt on the method’s empirical advantage. In particular, it is counterintuitive that the PI-based posterior sampling variant outperforms the EI-based one, despite the theoretical motivation.
>
> We believe that the performance of GP-PIMS and GP-EIMS in our experiments is almost the same.
> We would like to claim that GP-EIMS achieves both sublinear BCR upper bounds and good optimization performance without any practical heuristics, especially compared with other theoretically guaranteed EI-based methods such as EI-$\mu^{\rm max}$.
> We have revised the manuscript to clarify the descriptions claiming the empirical advantage of GP-EIMS in Sections 1, 5, and 6.
>
> On the other hand, the difference between GP-PIMS and GP-EIMS is of interest.
> GP-PIMS achieves the slightly tighter BCR upper bounds compared with those of GP-EIMS.
> GP-EIMS is more explorative compared with GP-PIMS due to the definition of AF.
> That is, for the same training dataset, the posterior variance chosen by GP-EIMS is larger than that of GP-PIMS.
> Hence, although both GP-PIMS and GP-EIMS are guaranteed in terms of regret, their balances for the exploitation-exploration trade-off are different.
> Therefore, we believe that there are cases where GP-EIMS is superior to GP-PIMS, and vice versa, due to the exploitation-exploration trade-off.
>
> >Additionally, stronger baselines such as Joint Entropy Search are not considered, further weakening the empirical positioning.
>
> We have added E$^3$I and JES as baselines.
> We observed that these two methods exhibit superior performance in terms of cumulative regret.
> However, their simple regrets sometimes stagnate.
> These results suggest that E$^3$I and JES can sometimes lead to over-exploitation in our experiments.
>
> >Comparison with $E^3I$: one important aspect, since this-EI variant can be regarded as a degenrate version of $E^3I$, it is important to understand its empirical performance compared with it. It is not expected that EIMC is empirically better than $E^3I$, but this provide a better position of this paper in terms of its empirical performance.
>
> As discussed above, we observed that GP-EIMS is often superior to E$^3$I regarding the simple regret.
> We conjecture that E$^3$I is too exploitative compared with GP-EIMS in our experiments for the following reasons.
> First, showing the relatively low cumulative regret and large simple regret implies that the corresponding BO method repeatedly evaluates non-optimal points whose instantaneous regret is low but not zero.
> Second, GP-EIMS can randomly show both exploitative and explorative behavior.
> That is, if $g^\star\_t$ is smaller, GP-EIMS becomes more exploitative, and vice versa.
> On the other hand, since E$^3$I takes the average of the GP-EIMS acquisition function, its behavior is more stable.
> In particular, if $g^\star\_t$ is smaller, then the GP-EIMS acquisition function value becomes larger.
> Therefore, E$^3$I is more strongly affected by the small $g^\star\_t$ samples than the large $g^\star\_t$ samples, which makes E$^3$I more exploitative.
> Hence, we can conjecture that E$^3$I results in over-exploitation not only from the experimental results but also from a mathematical interpretation.
>
> >Clarification of randomness in expectations: The statement "the expectation is taken with all the randomness, that is, $f$, $\epsilon$, and the randomness of the algorithm'' should be clarified. Specifically, the ``randomness of the algorithm'' refers only to the sampling of $g_t$ from the posterior GP, i.e., $g\_t \sim \mathcal{G} \mathcal{P}(\mu\_{t-1}, k\_{t-1})$. Otherwise, this may be misleading.
>
> Here, we provided a general description of the BCR.
> Therefore, the randomness of the algorithm may imply the random variable except for the posterior sampling.
> For example, IRGP-UCB performs the sampling of the confidence parameter $\beta_t$.
> Therefore, we have added the description stating that, in GP-TS, GP-PIMS, and GP-EIMS, the randomness of the algorithm refers to the posterior sampling.
>
> >Inaccurate citation?: The citation ``Kandasamy et al., 2018'' appears to focus on Bayesian simple regret rather than Bayesian Cumulative Regret (BCR). It would be more accurate to either remove this citation or clarify the distinction.
>
> We have removed the citation of Kandasamy et al. (2018) right before the BCR definition.
>
> >Justification for BCR focus: For accessibility to a broader audience, it is recommended to include a sentence justifying the choice of BCR as the analysis objective, instead of the more commonly used high-probability cumulative regret bounds (e.g., those in GP-UCB)
>
> We have added descriptions in response to the reviewer's comment.

---

> ### Author Response · Authors · 2025-08-22
>
> >Improved proof accessibility: While the paper is mostly clearly written, the regret proof in Appendix C remains difficult to follow without substantial background. i strongly recommend including a proof sketch or a high-level discussion at the beginning of the theoretical section. This would help guide readers by outlining the logical structure of the proof.
>
> We have added the explanation of the proof flow in the first paragraph of Appendix~C.
>
> >Since it is assumed that hyperparameters are known within this research, could the author provide some comments on how the regret bound degrades under GP model misspecification (e.g., mismatched kernel, unknown hyperparameters)?
>
> To our knowledge, in the frequentist setting where $f$ is an element of some RKHS, $\Omega(\epsilon T)$ regret lower bound has been shown [1].
> On the other hand, in the Bayesian setting, roughly speaking, if the ratio $p(f) / q(f)$ between the true unknown prior $p(f)$ and the prior used in the actual algorithm $q(f)$ is bounded from above as $p(f) / q(f) \leq C, \forall f$, then $\mathbb{E}\_{p(f)} [f(x^\star) - f(x\_t)] \leq C \mathbb{E}\_{q(f)} [f(x^\star) - f(x\_t)]$ [Sec 3.1 of 2].
> Therefore, if $p(f) / q(f) \leq C, \forall f$, the deterioration caused by the misspecification is only a constant term.
> If the above ratio can be infinity, we conjecture that $\Omega(\epsilon T)$ regret lower bound can be obtained as with the frequentist setting.
>
> We conjecture that a similar discussion holds for the case where hyperparameters are unknown and the BO algorithm uses wrong hyperparameters.
> On the other hand, if the hyperparameters are chosen by some approach, such as [3, 4], we may obtain a sublinear regret upper bound under several assumptions.
>
> - [1] Ilija Bogunovic, Andreas Krause, Misspecified Gaussian Process Bandit Optimization, Advances in Neural Information Processing Systems 34 (NeurIPS 2021)
>
> - [2] Daniel Russo, Benjamin Van Roy (2014) Learning to Optimize via Posterior Sampling. Mathematics of Operations Research 39(4):1221-1243.
>
> - [3] Ziyu Wang, Nando de Freitas (2014) Theoretical Analysis of Bayesian Optimisation with Unknown Gaussian Process Hyper-Parameters, arXiv:1406.7758
>
> - [4] Juliusz Ziomek, Masaki Adachi, Michael A. Osborne, Bayesian Optimisation with Unknown Hyperparameters: Regret Bounds Logarithmically Closer to Optimal, Advances in Neural Information Processing Systems 37 (NeurIPS 2024)

---

### Review · Reviewer_5GpT · 2025-08-09

**Summary Of Contributions:**

The paper proposes a new algorithm for a Bayesian optimization setting where the objective function is drawn randomly from a centered Gaussian Process, and the observation noise is assumed to be Gaussian with a known variance. The new algorithm is a variant of the expected improvement framework, in which the reference value is chosen by maximizing a posterior sample path. Two sub-linear regret bounds for finite and continuous domain are provided. The analysis and proof techniques are mostly leveraged from existing works.

**Additional Comments:**

- Is there a known lower bound for BCR?

**Audience:**

Yes

**Audience Explanation:**

The new algorithm and its sublinear BCR bound are novel, and likely of interest to the Bayesian optimization community.

**Claims And Evidence:**

Yes

**Claims Explanation:**

The paper's main contributions are Algorithm 1 and its regret bounds in two theorems 4.6 and 4.8. I carefully verified the proofs of these theorems in Appendix C and found all proofs to be correct. I also verified the claim made in Appendix A on the mistake of an existing paper.

**Requested Changes:**

Main:
- It would be better to explicitly point out exactly what the novelty in Algorithm 1 is in comparison to previously known algorithms in existing works. In particular, is using $g_t^*$ in the EI function the main novelty of Algorithm 1?
- In Lemma D.2: Is $E[\max_x f(x)] = E[\min_x f(x)]$ a typo and it should be $E[\max_x f(x)] = -E[\min_x f(x)]$?

Minor:
- In the Introduction, it would be helpful to be more explicit about the role of the Acquisition Function in this framework. For example, writing $x_t \in \argmax AF(x)$ would be good.
- In the first sentence after Equation 165: why does the monotonicity of $\sigma_t$ implies $\sigma_{t-1}(x) \leq 1$? I thought this came from the normalized kernel $k(x,x) \leq 1$.
- Right before Equation 175: it is helpful to remind the reader that $g_t^* = g_t(z_t)$ here.
- The sentence right after Equation 178: inside the word "distribution" there is some Latex symbol, which seems like a compilation error.
- Going from Equation 179 to Equation 184: this part is correct, however can we simply take the integral from 0 to 1 w.r.t $\delta$ of $\log(\dots) + \beta(\delta) + \sqrt{2\pi \beta(\delta)}$ to get Equation 184? This is often done, for example, in bandit literature to convert a high-probability regret bound to an in-expectation regret bound.

---

> ### Author Response · Authors · 2025-08-22
>
> We appreciate the detailed and constructive comments.
> We have revised the paper with the updated parts highlighted in red font.
>
> >It would be better to explicitly point out exactly what the novelty in Algorithm 1 is in comparison to previously known algorithms in existing works. In particular, is using $g^\star\_t$ in the EI function the main novelty of Algorithm 1?
>
> Yes, using $g^\star\_t$ is the difference from the original GP-EI algorithm, although the E$^3$I algorithm uses samples from $p\_t (g^\star\_t)$ too.
> The difference from the E$^3$I algorithm is that GP-EIMS uses only one sample, in contrast to using many Monte Carlo samples in the E$^3$I algorithm.
> We have added the description stating the difference from the original GP-EI algorithm in Section~4.
>
> >In Lemma D.2: Is $E[\max f(x)] = E[\min f(x)]$ a typo and it should be $E[\max f(x)] = - E[\min f(x)]$?
>
> We thank pointing out our typographical errors.
> We have fixed this.
>
> >In the Introduction, it would be helpful to be more explicit about the role of the Acquisition Function in this framework. For example, writing $x\_t = {\rm argmax}\ AF(x)$ would be good.
>
> We have added the suggested description in Sections~1 and 2.
>
> >In the first sentence after Equation 165: why does the monotonicity of $\sigma\_t$ implies $\sigma\_t(x) \leq 1$? I thought this came from the normalized kernel $k(x, x) \leq 1$.
>
> The reviewer's comment is correct.
> We have added the description after Eq. (165) and the assumption of $k(x, x) \leq 1$ in Lemma 4.2 and Corollary 4.5.
>
> >Right before Equation 175: it is helpful to remind the reader that $g^*\_t = g\_t(z\_t)$ here.
>
> We have added the suggested explanation around Eq.~(175).
>
> >The sentence right after Equation 178: inside the word "distribution" there is some Latex symbol, which seems like a compilation error.
>
> We were unable to find such a compilation error. We sincerely ask where the error is.
>
> >Going from Equation 179 to Equation 184: this part is correct, however can we simply take the integral from 0 to 1 w.r.t $\delta$ of $\log(\cdots) + \beta(\delta) + \sqrt{2\pi \beta(\delta)}$ to get Equation 184? This is often done, for example, in bandit literature to convert a high-probability regret bound to an in-expectation regret bound.
>
> We believe that taking the integral from 0 to 1 and taking the expectation of $Z \sim {\rm Uni}(0, 1)$ are identical.
> Therefore, although we can rephrase it, the simplicity of the description remains unchanged.
>
> >Is there a known lower bound for BCR?
>
> To our knowledge, [1] is the only known lower bound in the Bayesian setting.
> However, since [1] shows the lower bound of the expected regret over noise (not averaged by Gaussian processes) for a one-dimensional input domain, we believe that a more general BCR lower bound has not been shown.
>
> [1] Jonathan Scarlett. Tight regret bounds for Bayesian optimization in one dimension. In Proceedings of
> the 35th International Conference on Machine Learning, volume 80 of Proceedings of Machine Learning
> Research, pp. 4500–4508. PMLR, 2018

---

> > ### Comment · Reviewer_5GpT · 2025-08-22
> >
> > Thanks for the convincing answers. The edits look good to me.
> >
> > I have several additional (minor) comments below.
> >
> > - About the typo $E[\max f(x)] = E[\min f(x)]$: there is still one such typo in the first sentence in the proof of Lemma D.2.
> > - I did not carefully verify the proofs in Appendix B. This is mostly due to my limited capacity, but also because it is not clear to me how the theoretical results in Appendix B impact the main theorems and/or key takeaways of the paper.
> > - The ∵ symbol in place of "because" caught me by surprise, although it was not hard to figure out what that means. In my opinion, it is more reader-friendly to spell out the text "because" instead of that symbol, or if space/margin is an issue then "as" would suffice. Of course, this is mostly a style thing - maybe it is common within the Bayesian optimization community, so feel free to ignore this comment.

---

> > > ### Author Response · Authors · 2025-08-26
> > >
> > > We appreciate the constructive comments.
> > >
> > > >About the typo $\mathbb{E}[\max f(x)] = \mathbb{E}[\min f(x)]$: there is still one such typo in the first sentence in the proof of Lemma D.2.
> > >
> > > We are sorry for our carelessness.
> > > We have fixed it in the revised paper.
> > >
> > > >I did not carefully verify the proofs in Appendix B. This is mostly due to my limited capacity, but also because it is not clear to me how the theoretical results in Appendix B impact the main theorems and/or key takeaways of the paper.
> > >
> > > The novelty of the proofs in Appendix B is limited to (i) adaptation of the analyses by Wang and Freitas (2014) for the frequentist setting to the Bayesian setting and (ii) the cumulative regret analysis for the variant of GP-EI using ${\rm max}\_{i \in [t-1]} \mu\_{t-1}(x\_i)$ (Trant-the et al., 2022).
> > > These analyses are almost straightforward from (Wang and Freitas 2014, Tran-the et al., 2022) but have not been shown explicitly to our knowledge.
> > > Therefore, we have shown them in Appendix B for a more comprehensive discussion.
> > > We believe that these analyses are helpful to readers in the BO community.
> > >
> > > >The $\because$ symbol in place of "because" caught me by surprise, although it was not hard to figure out what that means. In my opinion, it is more reader-friendly to spell out the text "because" instead of that symbol, or if space/margin is an issue then "as" would suffice. Of course, this is mostly a style thing - maybe it is common within the Bayesian optimization community, so feel free to ignore this comment.
> > >
> > > We have revised the paper as the reviewer suggested to improve clarity for a broad audience.
> > > For descriptions that are not in sentence form, we have rephrased them using phrases such as "because of," e.g., "because of Jensen's inequality."

---

### Decision · Action_Editor_Yk7a · 2025-09-22

**Recommendation:** Accept as is

**Audience:**

Yes

**Audience Explanation:**

The central topic of this paper -- Bayesian optimization -- is of interest to a nontrivial subset of TMLR's audience. As the authors propose and study a novel algorithm in this space, there would certainly be interest from the TMLR community in the findings of this paper.

**Claims And Evidence:**

Yes

**Claims Explanation:**

This manuscript proposes and studies a novel acquisition strategy for Bayesian optimization (where the underlying model class is Gaussian processes) wherein we apply the common expected improvement (EI) acquisition function but using a reference value that is generated by maximizing a posterior sample path of the underlying GP. The authors both provide a nontrivial regret bound for this proposed algorithm and evaluate its performance in an empirical study, where it appears to work well.

Overall, this work was received well by the reviewers, who generally found:

- the paper of interest to the Bayesian optimization community (see below),
- the proposed algorithm well-motivated,
- the theoretical results to be sound, and
- the empirical study to be insightful.

The reviewers did have a few minor concerns regarding the some details; however, the discussion period with the authors was sufficient to overcome these concerns. Ultimately, the reviewers were unanimous in their recommendation to accept the manuscript (after revision by the authors during the discussion period) to TMLR.